

# Sediment loading in Fennoscandia during the last glacial cycle

Wouter van der Wal[1], Thijs IJpelaar[2]

[1]Faculty of Aerospace Engineering, Delft University of Technology, Delft, 2613 DH, Netherlands
[2]'s Hertogenbosch, 5211 TL, Netherlands

*Correspondence to*: Wouter van der Wal (w.vanderwal@tudelft.nl)

## Abstract

Models for postglacial rebound, or Glacial Isostatic Adjustment (GIA) routinely include the effects of meltwater redistribution and changes in topography and coastlines. Since the sediment transport related to the dynamics of ice sheets

may be comparable to that of sea level rise in terms of surface pressure, the loading effect of sediment deposition could cause measurable ongoing viscous readjustment. Here we study the loading effect of glacial induced sediment redistribution (GISR) related to the Weichselian ice sheet in Fennoscandia and the Barents Sea. The surface loading effect and its effect on the gravitational potential is modelled by including changes in sediment thickness in the sea level equation following the method of Dalca et al. (2013). Sediment displacement estimates are estimated in two different ways: (i) from a compilation

of studies on smaller features: through mouth fans, large scale failure and basin flux and (ii) from output of a coupled ice-sediment model. To account for uncertainty in Earth's rheology three viscosity profiles are used.

It is found that sediment transport can lead to changes in relative sea level of at most several meters with the largest effect occurring earlier in the deglaciation. This magnitude is below the error level of most of the relative sea level data because data are sparse and errors are larger for older data. The maximum effect on present-day uplift rates is a few tenths of mm yr⁻

¹, which is around the measurement error of long-term GPS monitoring networks. The maximum effect on present-day gravity rates as measured by the GRACE satellite mission is up to tenths of $\mu$Gal yr$^{-1}$, which is larger than the measurement error.  Since GISR causes systematic uplift in most mainland Scandinavia, including GISR in GIA models would improve interpretation of GPS and GRACE observations there.

## 1. Introduction

Erosion in glaciated areas is larger than in non-glaciated regions (Hallet et al., 1996), and estimates for sediment deposition in glaciated regions vary from millimeters per year to centimeters per year close to glaciers (Elverhoi 1984; Finlayson 2012), which is comparable to changes in sea level during the last glacial cycle. The loading effects of meltwater redistribution are routinely included in models of Glacial Isostatic Adjustment (GIA), but the loading effect of sediment transport is not. Viscoelastic relaxation due to sediment deposition has been shown known to cause present-day subsidence rates in the

Arabian Sea and the Indus River basin (Ferrier et al., 2014), and also the Gulf of Mexico (Ivins et al., 2007; Simms et al.,



2013) although a recent estimate reduces that amount to 0.5 mm yr$^{-1}$ (Wolstencroft et al., 2014). With the large amount of sediment transport involved in glacier growth and melt it is possible that some of the present-day observed vertical motion near previously glaciated areas is caused by past glacial induced sediment redistribution (GISR) rather than changes in ice or water load in the last glacial cycle. The present-day observations are used to invert for viscosity or for ice thickness in GIA

models which means those inferences could be biased when GISR is not taken into account. The objective of this paper is to find out what is the effect of GISR during the Weichselian on GIA observables in Fennoscandia including the Barents Sea. The interest in this region stems from the fact that glacigenic sediment transport is large there (Elverhøi, 1984). The last 2,5 million years of glacial erosion resulted in a sediment layer of several kilometre thickness (Riis and Fjeldskaar 1992) and several observations of sediment deposition are available (e.g. Dowdeswell et al., 1996) from which the loading can be

quantified. Here, the focus is on present-day uplift, gravity rates and relative sea level data, which are routinely used to constrain GIA models.

Models exist which couple the sediment displacement to the movement of glaciers (e.g. Boulton 1996; de Winter et al., 2012), but since the ice sheet thickness, as in most GIA models, is not a dynamic model component, erosion is not coupled to the changes in ice thickness in this study. Instead, the amount of GISR is derived from literature on observed sediment

deposits and reported output from a coupled ice-sediment model. Sediment being deposited in the ocean will not only induce vertical motion, but also displace water and affect the gravity field. To model this effect we use the methodology of Dalca et al., (2013) to include sediment redistribution in the sea-level equation in a self-consistent way. Dalca et al., (2013) show that the relative effect of time-varying ocean load resulting from sediment redistribution can be up to 40% in terms of relative sea level. The method will be discussed briefly in Sect. 2. After that, it is explained how different estimates of GISR are created.

Next, sea level change and uplift rates are calculated for the different sediment transport scenarios and conclusions are drawn about the relevance of GISR in explaining GIA observations.

## 2. Method

The loading effect of ice and meltwater are routinely included in GIA models. The so-called sea-level equation is solved, which computes the sea-level distribution that accompanies a change in ice volume and corresponding changes in the Earth's

shape and potential field (Farrell and Clark 1976; Mitrovica and Peltier 1991). The effect of sediments can also be included in the sea-level equation, as shown by Dalca et al., (2013). Here we follow Dalca et al., (2013), Kendall et al., (2005) and references therein. Only the key elements will be repeated here and some small differences will be pointed out.

The sea level equation is written as

$$SL = G - (R + H + I),\qquad\qquad\qquad\qquad\qquad\qquad\qquad\qquad\qquad(1)$$





where $G$ is the height of the equipotential surface coinciding with the sea level, $R$ is the height of the Earth's crust, $H$ is the thickness of sediments, $I$ is the thickness of ice masses supported by land. The total surface mass load is defined as the sum of the mass of water, ice and sediment:

$$L = \rho_w S + \rho_I I + \rho_H H ,\tag{2}$$

where $\rho_w$, $\rho_I$, $\rho_H$ are the respective densities. At each time step $j$ a check is performed to see whether ice is grounded or not by requiring that the ice starts to float when the pressure exerted by the ice (prescribed by the ice model) is equal to the pressure of the current sea level. Thus, floating occurs when

$$I_j < \left(SL_j + I_j\right)\frac{\rho_w}{\rho_I} .\tag{3}$$

Solving the sea-level equation requires iteration for each time step (the 'inner' iteration). The initial guess for the change in ocean height is given by

$$\delta S_j^{i=0} = \delta h_j C_j - T_p \left(C_j - C_{j-1}\right) - \delta H_j \tag{4}$$

Note that the sediment thickness is subtracted here because it is included in the definition of sea level.

A change in topography affects the location of coastlines and hence the distribution of melt water. At first the pre-glacial topography $T_0$ is assumed to be equal to the present-day topography $T_p$. After computing sea level increments at all time steps, this estimate can be improved using the total sea level rise:

$$T_0 = T_p + \Delta SL_p \tag{5}$$

With the improved pre-glacial topography, the computation of sea levels can be repeated (the 'outer' iteration) until the pre-glacial topography reaches convergence. Erosion will also change the topography, but this effect is not included; the effect of erosion on the location of coastlines is expected to be small.

The effect of sediment redistribution is implemented in the numerical codes for the sea-level equation developed by Schotman (2007). A partial benchmark against other numerical solutions of the sea-level equations was carried out in Spada et al. (2012). Rotational feedback is also included in the sea level equation following Wu and Peltier (1984) and Milne et al. (1998). The response of the Earth to surface loading for a radially symmetric Earth is computed with the normal mode method (Vermeersen and Sabadini, 1997) which is benchmarked in Spada et al. (2011).

## 3. Model inputs

The computation requires several inputs, such as elastic parameters and viscosity of the Earth the ice and sediment distributions, which are discussed in the following subsections. For the present day topography ETOPO5 is used. Note that





the computations of the sea level equation takes place in the spherical harmonic domain. The maximum spherical harmonic degree of 256 and size of the grid is 256x512.

### 3.1 Model inputs: viscosity and ice loading

In this study we consider a laterally homogeneous Earth model and vary the radial viscosity profile. As a reference model we use VM5a (Peltier and Drummond, 2008) which is an iteration of the VM2 profile model that is used in the creation of ICE-5G (Peltier 2004). As alternative models we select models that have been shown provide a good fit to sea level data, GPS and GRACE data in Fennoscandia in Root et al. (2015b). That study found two model sets, one with higher viscosities in upper and lower mantle viscosity and one with lower viscosities. Out of those sets we select the two models M8-128-150 and

M4-16-80, where the first number denotes the upper mantle viscosity in $10^{20}$ Pa s, the second number denotes the lower mantle viscosity in $10^{20}$ Pa s and the third number denotes the lithosphere thickness in km. The three viscosity profiles are shown in Fig. 1. Note the lower viscosity $10^{22}$ Pa s in VM5a just below the lithosphere, from 60 to 100 km depth.

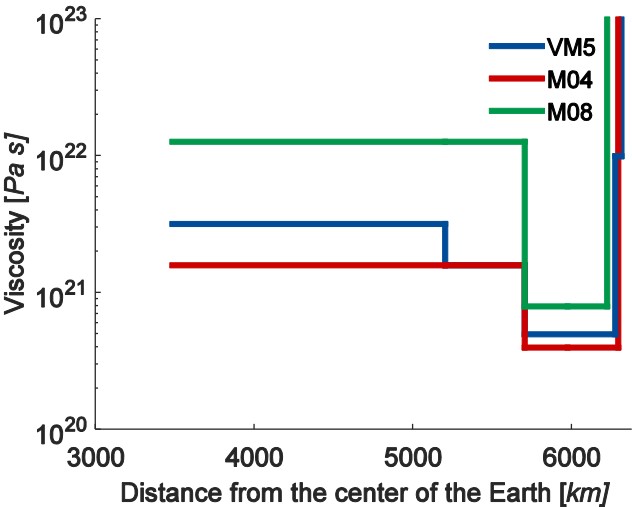

**Figure 1: The three viscosity profiles used in this study.**

Since we are only interested in the effect of GISR the exact ice loading history is of less importance, and only influences the effect of GISR through the distribution of meltwater possibly replaced by sediment, which is a small effect. For ice loading history the ICE-5G modelv1.2 (Peltier 2004) is selected. Time steps are 2,000 years from 120 kyr Before Present (B.P.) to 32 kyr B.P., 1,000 years from 32 to 17 kyr B.P. and 500 years from 17 kyr B.P. to present.



## 3.2 Model inputs: sediment distribution

In order to model the loading effect of GISR it needs to be known how much sediment is transported, where it came from and where it is deposited. Erosion and deposit estimates for entire Scandinavia during the last glacial cycle are not readily available. Therefore, we created a map of sediment deposition from estimates in the literature of sediment volumes transported in smaller features: through mouth fans (TMFs), large scale failures and basin flux. Each of the features will be briefly discussed in the following. TMFs are places where rapid flowing ice streams at the end of the continental shelf converge, and where sediment is deposited off the shelf, see Figure 2 for the locations. Local observations come from sonar and seismic profiling (Dowdeswell et al., 1996) and coring (Saettem et al., 1992; Laberg and Vorren 1996; Taylor et al., 2002; Laberg et al., 2012). Other estimates come from modelling based on bathymetry, elevation and environmental conditions (Siegert and Dowdeswell 2002), but these are highly dependent on the amount of ice that is believed to have existed in the Barents and Kara Sea. The estimates are compiled in Table A1 in Appendix A.

Large scale failures represent the sediment that is displaced after collapse of the slope. The largest of such events related to the Eurasian ice sheet is the Storegga slide. Haflidason et al. (2004; 2005) estimate the Holocene event to have a volume of 2400-3200 km$^3$ based on sonar scans and sedimentary cores. A compilation of the studies for this and other slides is provided in Table A2 in Appendix A.



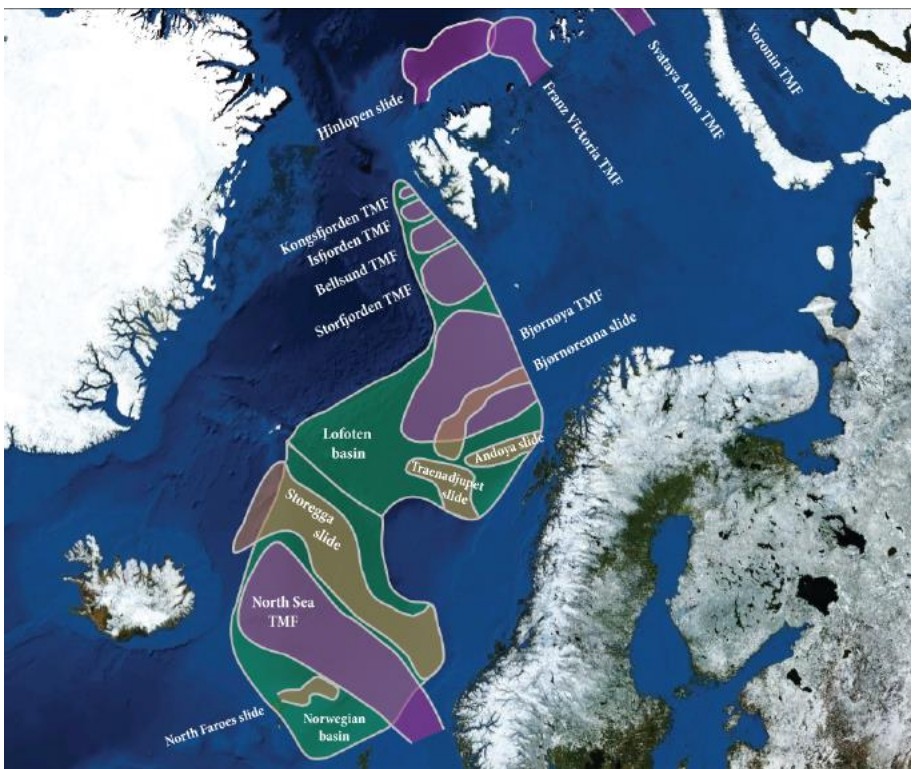

**Figure 2: Schematic representation of the locations of Through Mouth Fans (purple), Large scale failures (brown) and basins (green).**

The TMFs and the large-scale failure are examples of local features. However, constant deposition of sediment from the source area to the basins (Figure 3) also takes place. During glaciations, sediment activity is increased, with thickness changes estimated to be 5 cm kyr$^{-1}$ and 2 cm kyr$^{-1}$ for the periods of 30-10 kyr B.P. and 10-0 kyr B.P. respectively (Taylor et al., 2002), which amounts to a total volume of 97 and 58 km$^3$ for the Norwegian basin and 485 and 290 km$^3$ for the Lofoten basin. There are also channels and canyon systems that are not captured by either the large scale individual features which still provide larger sedimentation rates than the overall basin estimate. Estimates for the Lofoten channel system amount to 35 km$^3$ (Taylor et al., 2000), which is small enough that it can be neglected compared to the other events. The basin fluxes are given in Table A3 in Appendix A.

Conflicting estimates are stated for GISR volumes in Tables A1-A3 in Appendix A, for example the Byørnøya Through Mouth Fan in Table A1. Also the timing is uncertain for most events. Therefore, different sets of GISR loads are created, consisting of minimum, maximum and moderate estimates from the tables. The time step is chosen to be the average of the time span given, rounded off to the nearest time step of the ice model. To create the spatial distribution of sediment contours were drawn in the QGIS software package, to match the figures in Taylor et al. (2002) and Winkelmann et al. (2008), see Figure 3. The resulting shape files were converted to raster files with sediment height for each grid point using MATLAB



and GDAL. Across the source and sink areas of the sediments a uniform sediment height change is assumed so that the volume matches the estimates in Table B4. The areal extent is the largest source of uncertainty. To address this we looked for an alternative estimate of GISR.

The model of Amantov et al. (2011) couples ice sheet growth and erosion and is constrained by sedimentary and seismologic observations. Sediment removal and deposition is shown in Figure 4, which is figure 3.6 of Amantov et al. (2011). Because part of the data is proprietary, the data for the time series and also for creating the figure is not available. The software package QGIS was used to georeference the image and find the sediment thickness at hundreds of points between which triangular interpolation was performed. An average rock density of 2300 kg m$^{-3}$ was used to obtained sediment thickness values. The original model of Amantov et al. (2011) conserved mass, but in converting the graphics to a grid of sediment thickness values mass conservation was lost. We opted not to enforce mass conservation, as doing so would require further assumptions which introduce uncertainty. As a result there is a geoid shift of around 13 cm, which causes a loading effect that is small compared to sediment thickness changes. To obtain a time series of sediment thickness it is assumed that the temporal variation of sediment transport follows the total ice mass change, based on the fact that erosion rate is proportional to sliding speed, which is enhanced during melting (Herman et al., 2011).

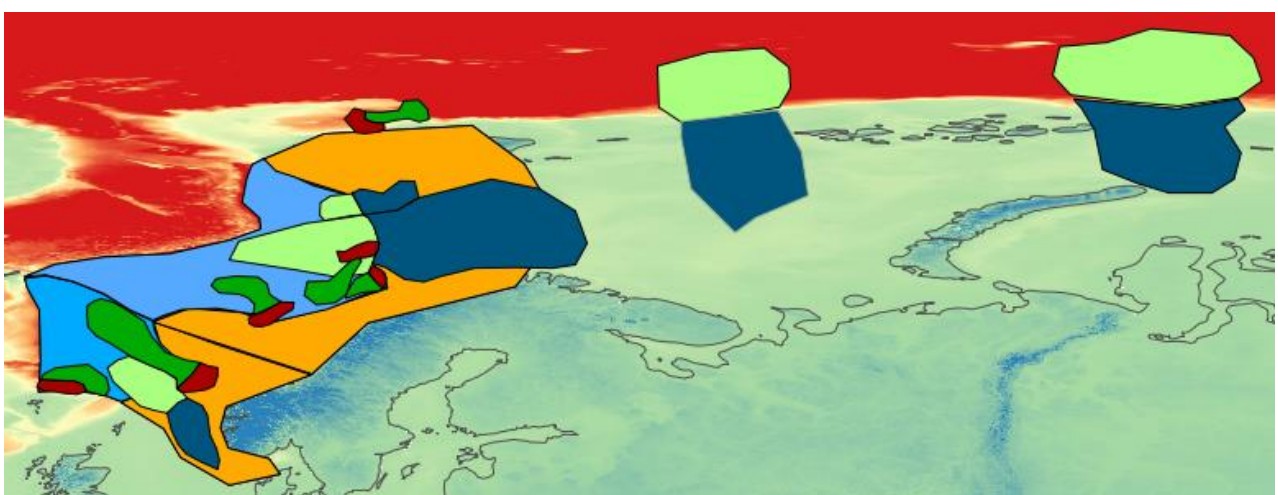

**Figure 3: location of erosion events, derived from Taylor et al. (2002) and Winkelmann et al. (2008).**




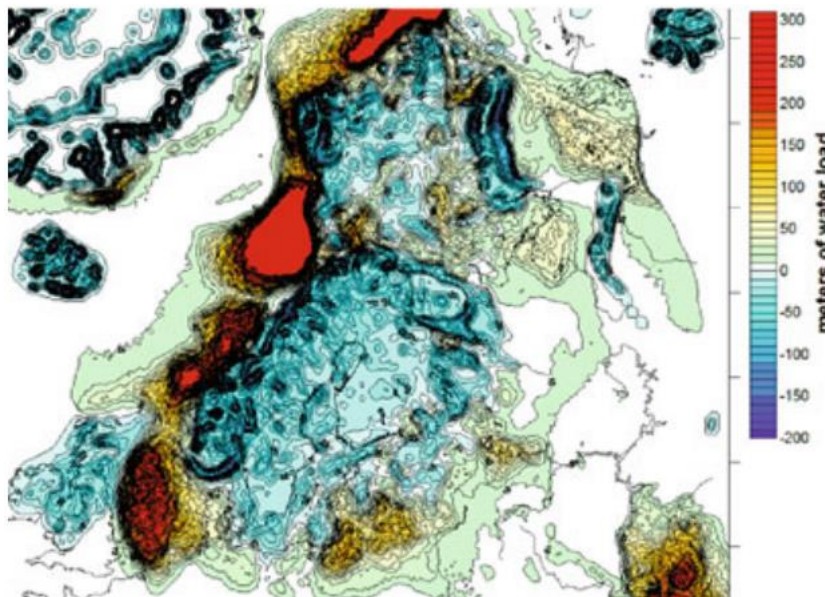

**Figure 4: Total Weichselian erosion and redistribution, figure copied from Amantov et al. (2011). The sediment load is converted to equivalent water thickness using average rock density. Note that the color scale is saturated at 300 m.**

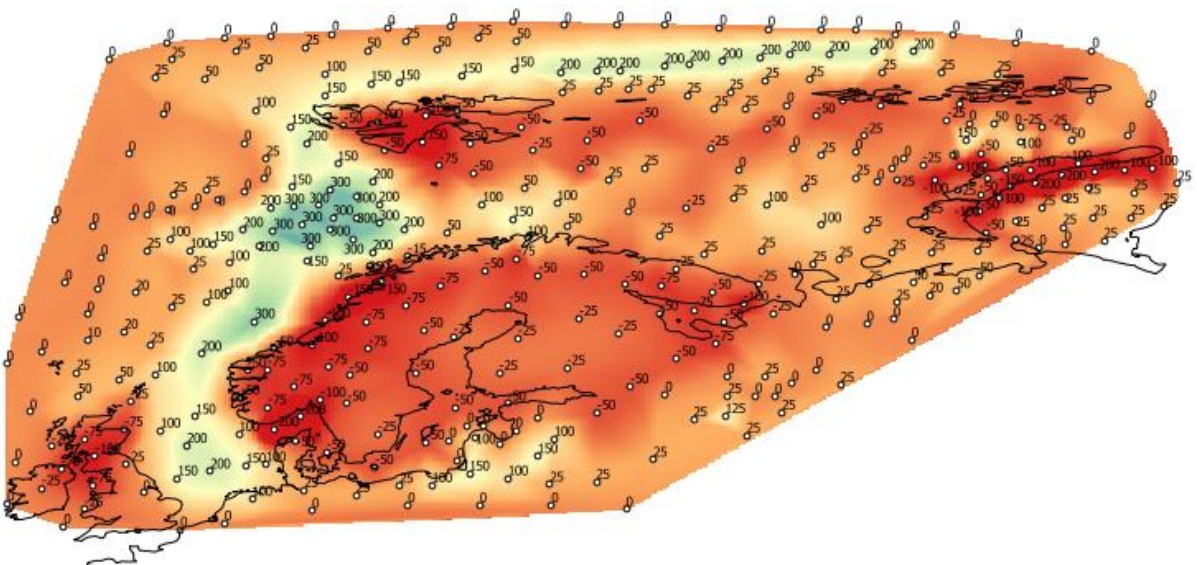

5   **Figure 5: Screen shot of points captured in QGIS from Figure 4.**

The color scale in Figure 4 is saturated at 300 m, and contour lines also stop there. To try and take into account that the distribution was higher at those locations, two models were created: one in which the cut-off of 300 m was maintained (Amantov1) , and one in which maximum thickness was set to 600 m and surrounding thicknesses interpolated from the edge



of the 300 m cut-off (Amantov2). The difference was small, but nevertheless results for both models will be used for the results. Finally, large-scale failures, which are not included in the model of Amantov et al. (2011), are added from Table A2.

## 4. Results

Figure 6 shows the displacement for sediment model M4-16-80 together with a selection of RSL sites from the Tushingham and Peltier (1992) database. It can be seen that the largest subsidence caused by glacial sediment transport is off-shore, and the largest uplift is in the center of the continent. Hence the effect on coastal RSL data is limited. Another reason why the effect on RSL data is limited is that the GISR causes a difference in RSL that increases over time. Because sea level of current sites is set to zero, the largest difference occurs earlier in the deglaciation, which coincides with larger error bars on

the data, if records are available at all. This can be seen in Figure 7, which shows the effect on RSL with and without GISR for the ICE-5G model in combination with the M4-16-80 Earth model. Differences are at the level of several meters, which is below or near the error bar. Values for the other sediment and earth models are given in Table 1.

|  | Max. RSL [m] | Max. uplift rate [mm yr$^{-1}$] | Max. gravity rate Scandinavia [μGal yr$^{-1}$] | Max. gravity rate Barents Sea [μGal yr$^{-1}$] |
|---|---|---|---|---|
| **Sed1** | 1.5/0.9/1.6 | 0.14/0.07/0.16 | 0.019/0.009/0.021 | 0.004/0.004/0.014 |
| **Sed2** | 1.9/1.1/1.8 | 0.19/0.08/0.21 | 0.023/0.011/0.027 | 0.004/0.004/0.014 |
| **Sed3** | 1.3/0.7/1.3 | 0.11/0.05/0.12 | 0.014/0.007/0.016 | 0.002/0.002/0.005 |
| **Amantov1** | 4.1/2.5/4.5 | 0.38/0.28/0.47 | 0.030/0.052/-0.048 | 0.008/-0.016/0.014 |
| **Amantov2** | 5.0/2.6/4.5 | 0.38/0.28/0.47 | 0.032/0.052/-0.050 | 0.007/-0.018/0.014 |

**Table 1: maximum effect of sediment loading for different GISR estimates. In each cell the three numbers correspond to earth models M4-16-80/M8-128-150/VM5a, respectively. The maximum effect on relative sea level measurements is calculated as the maximum effect at any of the 6 sites of Figure 6 at the time at which there are measurements. The uplift rate is interpolated at the GPS sites of the BIFROST network presented in Lidberg et al. (2010) and the maximum is shown. The maximum positive gravity rate in Scandinavia is determined in the land area contained in the box with longitudes from 5° to 37° and latitudes from 55° N to**

**71° N. The maximum positive gravity rate in the Barents Sea is determined in the box with longitudes between 10° and 100° degrees, and latitudes between 71° N and 81° N.**





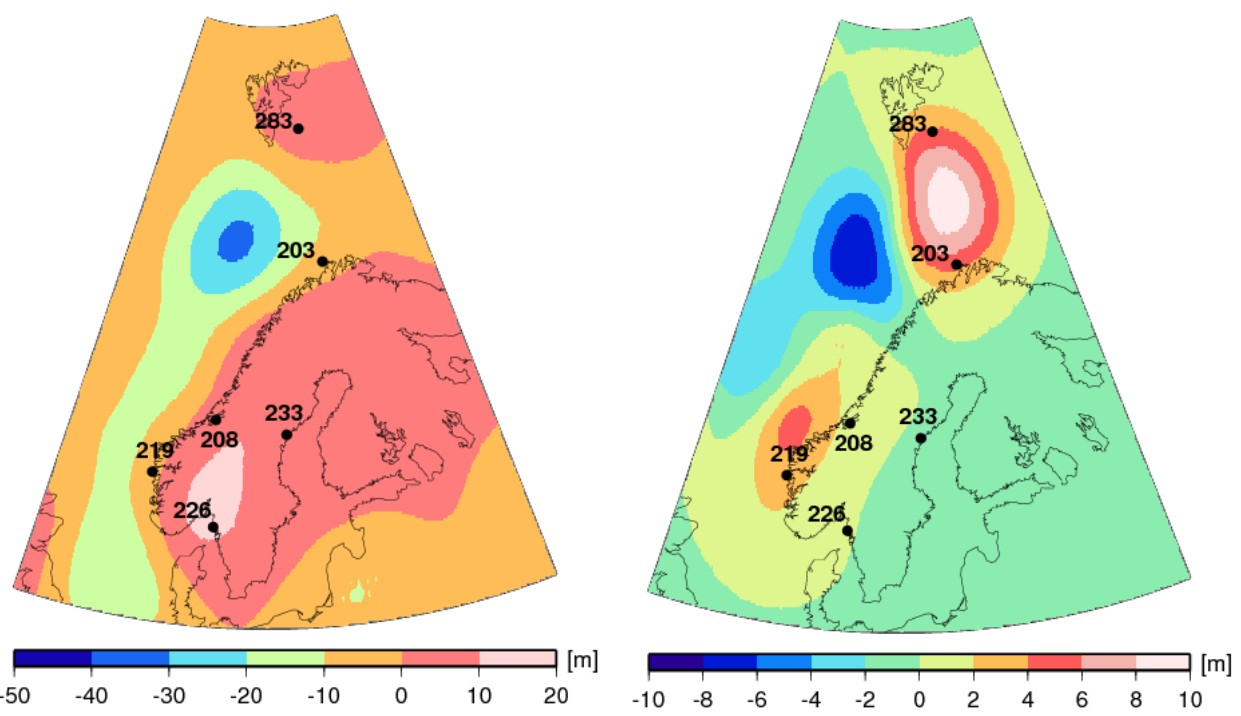

**Figure 6: Colors denote the difference between RSL at LGM and present caused by GISR for sediment model Amantov2 (left) and Sed1 (right). Note the different color scales. locations of Relative Sea Level data used in Fig. 7. Earth model M4-16-80 is used for both cases.**

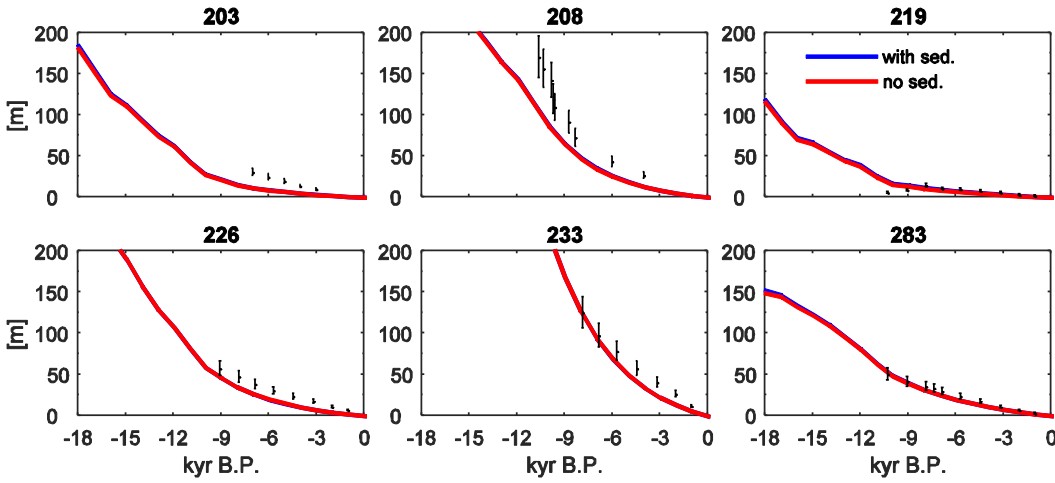

**Figure 7: RSL at selected sites from the Tushingham and Peltier (1992) database, for the ICE-5G model (red line) and the ICE-5G model with sediment transport (blue line) according to the Amantov2 model in combination with the M4-16-80 Earth model.**



While the relative sea level over time also includes the geoid effect due to removed mass, the present-day uplift rate only represents the viscous readjustment due to past changes in surface loadings. The pattern of uplift rates is shown in Figure 8 for the Amantov2 and Sed1 GISR models. In Figure 8a uplift can be seen in the formerly glaciated region of Scandinavia and subsidence off-shore, corresponding to Figure 5. Figure 8b mainly shows the effect of large scale failures. In both figures, the largest effects are off-shore where no GPS measurements can be made. To see the effect on observed uplift rates, the second data column in Table 1 shows the maximum effect of the sediment loading at any of the BIFROST sites of Lidberg et al. (2010). GISR is seen to always increase uplift rates, because most of the GPS measurement stations are in previously glaciated areas from where erosion took place. Thus, by interpreting the GPS rates as only resulting from ice unloading, the contribution from the ice is overestimated. This could result in biased inferences of ice thickness.

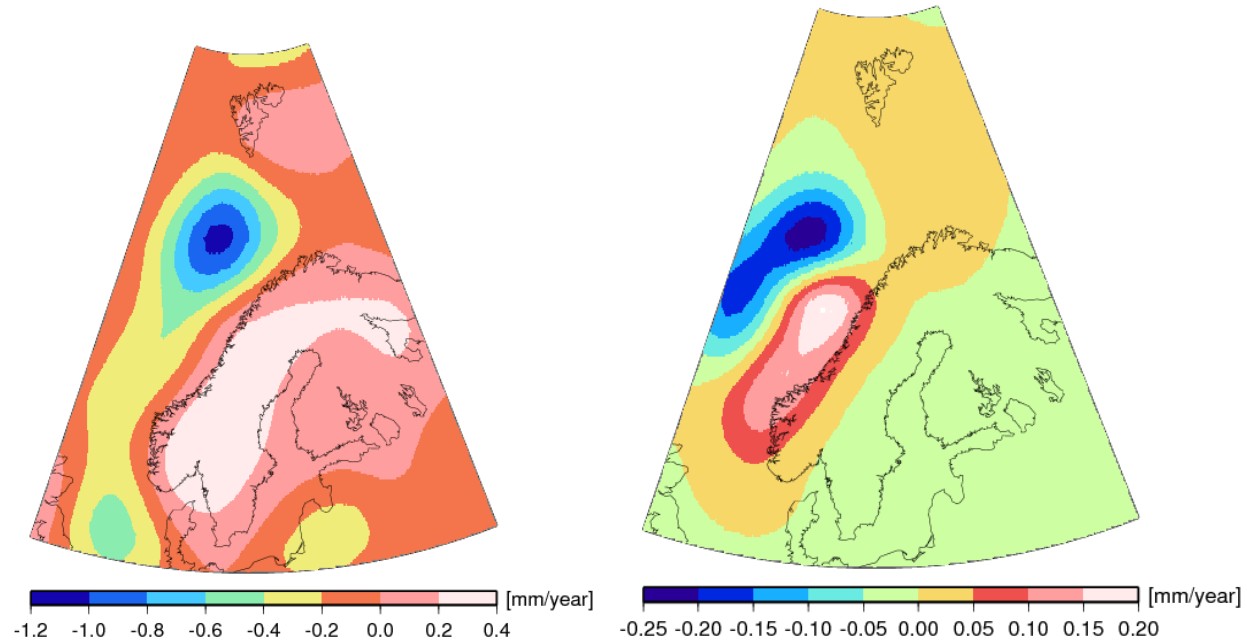

**Figure 8: Uplift rates caused by sediment redistribution according to GISR models Amantov2 (left) and Sed1 (right). Note the different color scales. Earth model M4-16-80 is used for both cases.**

Finally, the influence on gravity rates is also investigated, as gravity rates derived from the GRACE satellite mission constrain GIA in Scandinavia (Steffen et al., 2008; van der Wal et al., 2011) and the Barents Sea (Root et al., 2015a). To compare with GRACE data a maximum spherical harmonic degree of 60 is used in the GIA model, which is the same truncation used in many GRACE studies. Similar to the uplift rate signal, Sed1 reflects the signal of the landslides off-shore southern Norway and the Amantov1 model has negative gravity rates west of the Barents Sea where sediment is deposited. To evaluate the magnitude of the effect, Table 1 provides the maximum gravity rate that occurs in the areas that are used for GIA studies, Scandinavia and the Barents Sea, see the caption of Table 1 for how the areas are defined. The values can be



judged by comparing against the GRACE measurements error. The measurement error of the gravity rates derived from GRACE is computed using the method of Wahr et al. (2004), assuming that residuals after fitting a trend, secular and annual signal reflect noise. This method was shown to results in a similar error magnitude as calibrated standard deviations or a full variance-covariance matrix (van der Wal et al., 2010). The measurement error propagated to the trend has a value of 0.016

μGal yr$^{-1}$ for a ten-year GRACE time series from January 2003 to July 2013. In Scandinavia sediment loading results in gravity rates around the measurement error for the Sed/2/3 models for the three earth models, and larger effects for the Amantov1 model. This could affect the inference of a GIA signal from GRACE data in Scandinavia but less so in the Barents Sea where the GISR effect is smaller. Note that the current rate of sedimentation in the Barents Sea is not included in our computations. Sediment transport from the Barents Sea to the west would have the opposite effect on the gravity rate

but as of yet sediment transport is not yet detected in GRACE measurements.

## 5. Discussion and conclusions

We investigated the effect of sediment transport during the past glaciation in Scandinavia on current GIA observables. Although the amount of sediment transported is large, the effect is small compared to the ice loads that are displaced during glaciation, but comparable to water loading induced by ice sheet melt. Furthermore, sediment uptake takes place in a large

area, and deposition takes place in limited areas mostly confined to the ocean which can lead to locally higher signal. It was found that RSL data are not significantly affects by GISR because those data are located location near the shore, in between zones of erosion and deposition, and because a large part of the deposition takes place early in the deglaciation when the errors in RSL data are relatively large.

Also, the effect on present-day uplift rate and gravity rates is limited; depending on the estimate for sediment transport that is

20 used, the magnitude of GISR loading effects is near the measurement limit, several tenths of mm yr$^{-1}$ uplift rate and several tenths of μGal yr$^{-1}$ gravity rate. However the effect is systematic, reducing uplift rates and gravity rates in the land areas of Fennoscandia and Nova Zembla and increasing gravity rates west of Fennoscandia and the Barents Sea. Thus, if uplift or gravity rates are used to infer viscosity profiles or ice thickness those estimates could be biased.

Lateral variations in earth properties could affect the conclusions. The eastern part of Scandinavia is part of a craton, which

manifests in large crustal thickness and higher seismic velocities, which extend to the Barents Sea as seen in seismic measurements (see e.g. Schaeffer and Lebedev, 2013). The large seismic velocities likely result in high viscosity underneath eastern Scandinavia and the Barents Sea (e.g. van der Wal et al., 2013) which could reduce the uplift rate in East Scandinavia and the Barents Sea (Kaufmann and Wu, 1997), but could increase the uplift rate west of Norway.

To correct uplift and gravity rates for the effects of GISR it is necessary that more accurate estimates of sediment transport

are made, or that ice loading histories currently used in GIA models are coupled with erosion and sediment processes. We suggest to investigate the effect of GISR in other areas where last glacial ice caps were located close to the continental shelf and GISR is expected to be large, such as Antarctica and Alaska.



## Acknowledgements

We thank Riccardo Riva and Roland Klees for comments.

## Competing interests

The authors declare that they have no conflict of interest.

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



**Appendix A: estimates of sediment displacement**

| Through Mouth Fan | Volume [km³] | Timespan [ka B.P.] | Reference |
|---|---|---|---|
| Bellsund | -- | - | - |
| Byørnøya | 820-1100 | 22-17 | Taylor et al. (2002) |
| | 1360 | 21.5-17.5 | Laberg et al. (2012) |
| | 2700 | 30-8 | Siegert and Dowdeswell (2002) |
| | 4176 | Late Weichselian | Laberg and Vorren (1996) |
| | 4800 | 27-14 | Dowdeswell and Siegert (1999) |
| Franz Victoria | 500 | 30-8 | Siegert and Dowdeswell (2002) |
| Isfjorden | 22.5 | 30-0 | Elverhoi et al. (1995) |
| Kongsfjorden | - | - | - |
| North Sea | 800 | 30-0 | Taylor et al. (2002) |
| Storfjorden | 250 | 30-0 | Siegert and Dowdeswell (1999) |
| | 800 | 27-14 | Dowdeswell and Siegert (1999) |
| Svyataya Anna | Little | 30-8 | Siegert and Dowdeswell (1999) |
| | 2200 | 27-14 | Dowdeswell and Siegert (1999) |
| Voronin | little | 30-8 | Siegert and Dowdeswell (1999) |

**Table A2: estimates for volume of sediment displaced in different events, timespan and reference. No estimates for the Bellsund**
5  **and Kongsfjorden are available, but sediment transport there is expected to be relatively small.**

| Large Scale Failure | Volume [km³] | Timespan [ka B.P.] | Source |
|---|---|---|---|
| Andøya | 900 | 11-0 | Taylor et al. (2002) |
| Byørnøyenna | 1350 | 20-15 | Leynaud et al. (2009) |
| Hinlopen | 1200-1350 | 30 | Winkelmann et al. (2008) |
| North Faroes | 135-1700 | 9.85 | Taylor et al. (2002) |
| Storegga I | 3880 | 50-30 | Bugge et al. (1988) |
| Storegga II-III | 1700 | 8-6 | Bugge et al. (1988) |
| Storegga II-III | 2400-3200 | 7.25 | Haflidason et al. (2005) |
| Traenadjupet | 900-1900 | 4.2 | Taylor et al. (2002) |

**Table A3: Estimates for volume of sediment displaced by large scale failures.**





| | Rate [cm\ka] | Volume [km³] | Timespan [ka B.P.] | Reference |
|---|---|---|---|---|
| Norwegian basin | 2 | 58 | 10 | Taylor et al. (2002) |
| Lofoten basin | 2 | 97 | 10 | Taylor et al. (2002) |
| Norwegian basin | 5 | 290 | 20 | Taylor et al. (2002) |
| Lofoten basin | 5 | 485 | 20 | Taylor et al. (2002) |
| Norwegian channel | - | 35 | 30-0 | |

**Table A4: estimates for volume of sediment displaced in the Lofoten and Norwegian basins and channel systems.**

| | Sed3 | Sed1 | Sed2 | |
|---|---|---|---|---|
| **TMF** | **Volume [km³]** | **Volume [km³]** | **Volume [km³]** | **Time span [ka B.P.]** |
| Byørnøya | 820 | 1360 | 4800 | 22-17 |
| North Sea | 800 | 800 | 800 | 30-0 |
| Storfjorden | 250 | 300 | 800 | 27-14 |
| Franz Victoria | 500 | 500 | 500 | 30-8 |
| Svyataya Anna | 100 | 250 | 2200 | 27-14 |
| **Large Scale Failure** | **Volume [km³]** | **Volume [km³]** | **Volume [km³]** | **Time span [ka BP]** |
| Andøya | 900 | 900 | 900 | 8 |
| Bjørnøyrenna | 1350 | 1350 | 1350 | 18 |
| North Faroes | 135 | 1400 | 1700 | 10 |
| Storegga | 1700 | 2400 | 3200 | 7 |
| Traenadjupet | 900 | 1900 | 1900 | 4 |
| Hinlopen | 1200 | 1275 | 1350 | 30 |
| **Basin fluxes** | **Volume [km³]** | **Volume [km³]** | **Volume [km³]** | **Timespan [ka BP]** |
| Norwegian basin (interglacial) | 58 | 58 | 58 | 9-0 |
| Lofoten basin (interglacial) | 97 | 97 | 97 | 9-0 |





| | | | | |
|---|---|---|---|---|
| Norwegian basin (glacial) | 290 | 290 | 290 | 30-10 |
| Lofoten basin (glacial) | 485 | 485 | 485 | 30-10 |
| Channel | 35 | 35 | 35 | 30-0 |

**Table A5: sediment estimates created from observations and estimates of individual estimates in tables A1 to A3, as described in the main text. Sed2 and 3 are the maximum and minimum models, respectively.**