# Peer review of "Sediment loading in Fennoscandia during the last glacial cycle"

_Solid Earth, 2017_

## Referee Comment (RC1) · A. Purcell (Referee) · 20 Mar 2017

This paper presents a clear and very well presented discussion of the incorporation of sediment loading effects into calculations of sea level change and other observables related to GIA. I think this discussion is warranted and contributes to the accuracy with which GIA effects are calculated. I would suggest this paper be accepted with minor technical revisions which I identify below:

1) Page 1, Line 29: "shown known" should simply be "shown"

2) Page 2, Equation 1: This formula is mis-labelled and is incorrect. The quantity represented here is not sea-level it is water-depth. In this context $I$ should be multiplied by $\frac{\rho_i}{\rho_w}$ since it is not total ice thickness that is important, only that portion of ice thickness that displaces water.

3) Page 3, Equation 3: With the amendments suggested above the grounding line becomes $SL_j > 0$

4) Page 5, Line 3: "entire" should be deleted or replaced with "all of" or "the whole of"

5) Page 5, Line 7: "largest of such" should be "largest such"

6) Figure 3: The projection used for this figure makes the geography a little difficult to interpret.

7) Figure 4:It is very hard to make out the modern coastline which makes the figure difficult to interpret.

8) Figures 2, 3, 4, 5 and 6 use different projections. It would, I think, be better to standardise.

9) While there is a discussion of the uncertainty in the viscosity model and the observational record the uncertainty in the ice load seems to be the most significant element for this analysis. If GIA data are used to constrain the ice sheet and sediment changes are not considered then the change in ice thickness will be biased to compensate for the neglected sedimentary load. The implicit assumption that ice thickness is fixed and known is inaccurate.

---

## Referee Comment (RC2) · Anonymous Referee #2 · 28 Mar 2017

This study presents an analysis of sediment redistribution effects during the last glacial cycle on often used glacial isostatic adjustment (GIA) quantities. Such effects are in discussion for way more than a decade now and works by Dalca et al. (2013) and Ferrier et al. (2014) provided insight in the importance to incorporate this process in GIA modelling. This also shows the trend to upgrade specific models to advanced Earth System models. Van der Wal & IJpelaar present now results for Fennoscandia and the Barents Sea, an area well investigated in view of GIA, and show partly remarkable effects on relative sea-level, uplift and gravity change data. The authors developed 5 different sediment redistribution models and thus are also able to show some uncertainty of this effect.

The manuscript is well written in terms of structure and English, but some improvements should be done. Many are of rather technical nature, but should be addressed

carefully. I am willing to recommend publication after moderate revision.

Comments:

Title: as you deal with the Barents Sea, it should be included in the title. I'm reluctant to accept the term "sediment loading" in the title as you have erosional, thus unloading processes in your model as well. One can extent this discussion with the term "glacial-induced sediment redistribution" or "glacially induced sediment redistribution" used later in the manuscript. Are you sure that your model includes glacially induced processes only? What about the "normal" erosion-transport-sedimentation process during the last glacial cycle up to now? In some areas in your research area was no or only little glacially influenced redistribution of rock material, in other areas the normal rock material redistribution is altered by the glaciation, but one may question what is "normal" in this sense. What is in common is certainly that you are talking about the main exogene processes. Nonetheless, I would like you to think about the term you are using and use it already in the title. My suggestion would be "Effects of exogene processes in Fennoscandia and the Barents Sea during the last glacial cycle on glacial isostatic adjustment observations".

Abstract:

L8 remove "postglacial rebound, or"

L17 specify "at most several"

L20 change GPS -> GNSS and introduce abbreviation

L21 GRACE abbreviation unexplained

Main text:

P2L20 uplift rates of deformation and gravity change

Equation 1: what is SL?

Equation 2: what is L and S?

Equation 4: what is $h_j$ and $C_j$?

P3 second last: comma missing: Earth, the ice

Figure 1: switch x and y axes, then the figure is easier to get

Section 3.2: although dealing with longer time spans, it'd be good to mention the works by Zieba and co-authors (Zieba et al. 2016, 2017).

Figure 2: needs lat-lon information

P6L15: add Sed1, 2 and 3 to the min, max and moderate explanation.

P7L2: Table A4

Figures 3, 4 and 5 are awkward due to missing lat-lon info, different projection and more (Figure 3: why is it location of erosion events in the caption, but spatial distribution of sediment contours when referred to in the text? What are the colors? Name of areas? Why is there no contribution from land?; Figure 4: add information on blue and red, i.e. what is erosion what deposition. Amantov et al. (2011) seem to have a very fine resolution including major river systems, while Fig. 5 is rather coarse. Please explain in the text. Figure 5: color scale missing; figure not referenced in the text!) As these figures are of rather technical nature, I strongly suggest to move this part to the supplement, i.e. P6L16 to P9L2, and summarize briefly in 1-2 sentences that you developed in addition to sed1,2,3 two more models based on Amantov et al., and that further info how you got these models can be found in the appendix. Also mention briefly the differences between Amantov 1 and 2. To fill the space, I suggest to add a new Figure 3 showing a comparison of sed1,2,3 and Amantov1,2 similar as the old Figure 4, i.e. the total load change in metres of sediment thickness. Then you can explain this figure with the differences between the models.

P8L6: I understand that due to proprietary reasons Amantov's model is not available,

however, it should be possible to clarify via e-mail to Amantov or Willy Fjeldskaar if the saturation is at 300 m or any value above...

P9L2: Please mark in Table A2, e.g. with a star, which features were added to Amantov 1 and 2.

P9L7: it is rather the Oslo Graben area than the center of the continent where the largest uplift is found.

Table 1: comparing Sed1,2,3 with Amantov 1,2 it appears that the first underestimate GISR by mainly about a factor of 2 in view of the latter

P11L18: add "(not shown)"

P12L4: the value of 0.016 might be ok from the math but is from my perspective overly optimistic in this regard. We know meanwhile from GNSS that the value is something like 10+ mm/a. Depending on longer time spans, reference frame issues, better correction models for the atmosphere, tropospheric delay etc. the value may change, but not to 9.5 mm/a or less or to way more than 11 mm/a. Anything in between is well covered by the uncertainties. Analysing GRACE instead, you'll see that the maximum value can be 0.9, 1.2 or even 1.5 microGal/a depending on the solution centre, used filter, time span etc. This is not at all covered by such a small value. You need to specify this for the reader. In view of the max gravity rate you get (Table 1) the sediments loading effect appears to be negligible for now. Also, even if you use a cut-off of degree 60, GRACE data are still filtered and thus you should filter the GISR result as well then.

P12L16: affects -> affected; remove "location"

Table A4: add min, max, moderate to Sed1,2,3. Add data for Amantov1,2. What about the source areas? There is no information listed here nor in the main text. How are the continents treated? Please add!

Finally: will the models (at least Sed1,2,3) be available for download somewhere so that other can use them?

References

Zieba, K., Felix, M., Knies, J., 2016. The Pleistocene contribution to the net erosion and sedimentary conditions in the outer Bear Island Trough, western Barents Sea. arktos 2, 1-17.

Zieba, K., Omosanya, K.O., Knies, J., 2017. A flexural isostasy model for the Pleistocene evolution of the Barents Sea bathymetry. Norw. J. Geol. 97(1), 1-19.
* * *

---

## Referee Comment (RC3) · M. King (Referee) · 29 Mar 2017

The manuscript seeks to quantify the effect of post-LGM changes in sediment loading around Fennoscandia upon estimates of relative sea level, crustal motion and gravity changes. Sediment loading has only been addressed in a few papers, and this seems to be the first treatment for this region. The work is based on existing theory, although newly implemented within the authors' code. The conclusions are the effects of sediment loading are generally small, but may be important for present-day GPS and likely important for interpreting present-day gravity changes.

The paper was written in a way that suggested it was put together in a rush. The major changes I request below are largely related to figures and presentation. The English also needs some tightening up. I did not see any major flaws with the paper otherwise.

[Figure]

Major remarks: 1. Figure 3 is not a scientific figure. it is a perspective image, but cannot be interpreted. it needs colours scale, graticule, length-scale, axis labels etc.

2. Figure 4 is not appropriate. the authors have digitised this, so replace with the two variants of the reconstruction shown properly.

3. the authors use a sediment density of 2300 kg mˆ-3 but this choice is not defended or uncertainty tested. I presume sediment from different geological regions will have different densities.

4. I believe there is work on pro-glacial lake loading (Fleming et al?) that is possibly relevant to mention.

5. the authors present two models with sediment flux and other models that focus on offshore load changes, but do not present a sum of the two - it wasn't clear why

P1 L15 "smaller features" - smaller than what? (also later in the paper) L19: "older data" - older than what? vague. "the maximum effect..." needs some geographical context.

P2 L10: relative sea level does not go with "present-day" earlier in the sentence and so this needs a rewrite

P3 Eq 4 - lower case delta is used instead of upper I think. h and C are not defined. S should be SL

second last sentence : "parameters, the viscosity of the Earth, and the ice and ..." final sentence: how are marine grounded ice sheets handled - this is worth explicit mention given recent controversies (Purcell et al)

P4 L2: "is degree 256 and the ..." - not clear what the grid is L7/8: move "in Root ..." to "been shown by Root et al to. I note here these studies do not use sediment loading, so there's some issue here. L18: not clear if the timesteps refer to the input ice loading or something to do with the computation

[Figure]

P5: L3: "for the entirety of ..." L12: "Large scale ..." - sentence begins without context. This paragraph would benefit from the distance of the sediment transport being quantified. Figure 2 lacks any length-scale. it seems like a screenshot from Google Earth

P6L6: the sediment loading is taken to be up to present-day. Could the authors clarify how this change in rate corresponds with completion of deglaciation? surely the change in sediment loading scaled down as the ice sheet decayed? Or is some sediment transported by melt-water of subglacier outburst floods? L14: are -> were L16: add comma after "sediment"; remove next comma. There are no contours on Fig 3. L18: what is the resolution of the grid?

P7L7 find -> digitise. L8: reference FIg 5 here. L11: geoid shift - unclear what this means. over what period? L13: 'the fact' - really a fact? is it linearly proportional or non-linearly? how may that affect sediment transport rates?

Figure 5: suggest this can move to supplementary material. suggest you make a proper map in QGIS! it needs a colour scale

P9L2 results -> model input L5: M4-160-80 is not a sediment model. when is "displacement" relative to? Table 1: sed1, 2,3 are not referred to in the text and need descriptions. the box described in the caption could be shown on a figure

Figure 6 caption: clarify this is LGM to present loading changes Figure 7: suggest a second set of y-axis to show the difference between the two curves

P11L8: could the authors give an example of where people have used GPS to infer ice load? Figure 8: the locations of the Lidberg GPS shoudl be shown here or elsewhere

P12L2: residuals of what? L6: around the same magnitude? ;how much larger? L6-8: would be useful to understand typical % of signal L13: effect is small on what? L14: place over a large L15: higher signal than what? L16: affects->affected; delete location L18: suggest quantify the RSL effect here L22: not sure if these locations are all on
figures L26: in -> from L27: the *present-day* uplift

Matt King, March 29, 2017

---

## Referee Comment (RC4) · Anonymous Referee #4 · 30 Mar 2017

This study quantifies and presents the contribution of the sediments deposition in the GIA signal for the region of Fennoscandia for the first time. The authors implement a published methodology in their own codes and use sediment data retrieved from different (available) sources. Therefore the work is original and the results are sound. They clearly show that the sediment contribution for Fennoscandia is quite small and comparable to measurements error, but it is a potential source of systematic bias.

I do not have major concerns; however the English and the overall presentation can be certainly improved. In particular I found the motivation in the introduction poor and not clear. Consequently also the discussion (and conclusion) sounds just drafted. For example it is not clear at all if the result was expected or not. Some sentences suggest that the expected effect should be more important than what actually found, but there is no discussion about it at all.

The main motivation for this work is that the sediments deposition is expected to have a contribution comparable to the sea level feedback. However, on one hand the glacial erosion can build up kilometres of sediments in million years, so its absolute contribution is "large" (P 12, L 13 of the MS) but the deposition rate is actually small, usually few millimetres per year. On the other hand, the sea level feedback is a global effect that cannot be ignored, and the local sea level loading effect can be much larger than few millimetres per year. For example the retreating ice is replaced by the water, and if it is not correctly included the error is comparable with the ice loading effect. The difference between the effect of the sea level feedback and the sediment deposition should be addressed (in both introduction and discussion).

The authors note that in some other part of the world the sediment deposition has been proven to cause present-day subsidence. The state of the art, of those studies in particular, should be described in the introduction and comparison should be made in the discussion. Has that deposition occurred with higher rate? Is it more localised? Is there any similarity in those studies to the Fennoscandia sediment deposition? Or what are the differences?

Points that need to be cleared or rephrased:

P 1, L 25: "sediment deposition in glaciated regions vary from millimeters per year to centimeters per year, which is comparable to changes in sea level during the last glacial cycle". Do you mean the relative sea level? Do you mean local RSL in the same glaciated areas or in general the sea level effect in other regions?

P 2, L 7: "2,5 million years resulted in a sediment layer of several km thickness". This is misleading. For example 5 km in 2.5 million years is 2 mm/yr.

P 2, L 22: The Method section is not clear enough. Concepts defined in older studies are used here without properly recalling them. It is confusing even for people familiar with iterative method for solving the sea level equation. In Eq. 1 the SL variable (supposedly Sea Level) is not defined.

SED
P 3, L 2: In Eq. 2 the L is not defined. Add L: The total surface mass load "L" is defined as the sum...

P 3, L 5: "At each time step"... It is not declared that it is solved with a step-evolution.

P 3, L 9: "inner iteration" is not clear since you have not defined the method as iterative in a general way.

P 3, Eq. 4: C\_j is used without defining it.

P 3, L 18: Why "the effect of erosion on the location of coastlines is expected to be small"?

P 3, L 24: "The response of the Earth .... is computed with the normal mode method (Vermeersen and Sabadini, 1997) which is benchmarked in Spada et al. (2011)." Which code? Most codes benchmarked in Spada et al. (2011) implement the normal mode. And is therefore the code an incompressible model?

P 4, L 17: "... and only influences the effect of GISR through the distribution of meltwater possibly replaced by sediment, which is a small effect." What has a small effect? The melt water replaced by sediments? The fact that is small is not self-evident.

P 7, L 5: Is it so difficult to get Amantov data? And which is the original data accuracy?

P 8 Figure 4: I don't think Figure 4 is really necessary. You can easily indicate in Fig. 5 the areas where the colour scale from Amantov picture is saturated. So you spare a picture and the issue of copy rights.

P 8 Figure 5: This picture needs a colour scale bar.

P 10, Fig 7. RSL curves: the difference is not visible at all. The authors could make the red dashed. Since there's no visible difference this picture doesn't give more info than what you can tell in the text

P 11, L 8: "Thus, by interpreting the GPS rates as only resulting from ice unloading,

SED
the contribution from the ice is overestimated. This could result in biased inferences of ice thickness." This is one of the possible relevant effects and it should be (at least roughly) quantified. Is it a 1% overestimate or is it a 10%? However considering that most contribution of sediments to the uplift is in the sea and it is within the error, other source of error (such as the GPS reference frame error) could cancel it.

P 11, L 19: "To evaluate the magnitude of the effect, Table 1 provides the maximum gravity rate that occurs in the areas that are used for GIA studies, Scandinavia and the Barents Sea" Why not showing this with a picture?

P 12, L 13: "Although the amount of sediment transported is LARGE, the effect is small compared to the ice loads that" Large compared to ...? This sentence needs to be rephrased and this small effect should be explained (or discussed better). It's not clear if such small effect was expected or not. From the introduction I would guess that a larger effect was expected. So what is the main cause for a "large" amount of sediments to produce a "small effect"?

Minor comments

P 1, L 29: "has been shown known" -> shown or known?

P 2, L 7: "2,5" -> "2.5"

P 2, L 12: "Models exist which couple ...." does not sound like good English

P 2 , L 23: "The so-called sea-level equation is solved, which computes  $\ldots$  " does not sound like good English

P 4, L 9 and Figure 1: "M8-128-150 and M4-16-80" are these M04 and M08 in the figure? The names in the legend are not fully self-explanatory.

P 4, L 18: "modelv1.2" what is this? I understand that is a sort of update of the ICE-5G, is there a link or more detail to find that? What's the difference from the original?

P 6, L 17: Why citing QGIS and MATLAB, GDAL, which are only tool that implement
algorithm, and not instead citing the specific algorithms used?

P 9 Table 1: This table is not immediate to read. nn/nn/nn is visually not effective. Spaces could help, for example nn / nn / nn or even using 3 sub columns for each main column.

P 12, L 9: "Sediment transport from the Barents Sea to the west would have the opposite effect on the gravity rate but as of yet sediment transport is not yet detected in GRACE measurements." This sentence must be rephrased, I had to read it three times to understand it.

P 12, L 13: "ice loads" -> "ice loads effect"..

P 12, L16: "RSL data are not significantly affects" -> "RSL data are not significantly affected" and "because those data are located location near..." -> "because those data are located near..."

SED

---

## Referee Comment (RC5) · Anonymous Referee #5 · 5 Apr 2017

The main purpose of this study is to quantify sea-level responses to sediment redistribution caused by ice sheets in Fennoscandia over the last glacial cycle. To do so, the authors apply a recent sea-level model (Dalca et al., 2013), which computes sealevel responses to sediment erosion and deposition. The main finding is contained in Figure 7, which shows that sea-level responses to sediment redistribution are small in this region, such that accounting for sediment redistribution does not significantly help resolve differences between observed and modeled relative sea-level histories. This is a useful finding and the main strength of this study.

The manuscript has several weaknesses that I suggest the authors address before publication, most of which have to do with the presentation of the material. As I describe below, a number of items in the text are unclear, and most of the figures require major modification before they can be understood, particularly Figures 3-5. I do not

have major scientific concerns about the study, but two minor concerns are that the study did not conserve sediment mass, and it's not clear how eroded material was spatially distributed, which would make it difficult to reproduce the results of this study. I suggest the authors expand on these points in the text. Overall, I suggest that this study will be of interest to a number of readers in Solid Earth after major revision.

Additional comments

Page 1, line 17: I suggest specifying the timescale over which changes in relative sea level can be as large as several meters. Is this the integrated sea-level change from the Last Interglacial to the present?

Page 1, line 25: I suggest rephrasing this sentence, since glacial erosion is not always faster than non-glacial erosion. Glaciers frozen to their beds, for example, can inhibit erosion, rather than accelerating erosion.

Page 2, line 1: Does "that amount" in this sentence refer to subsidence rates due to sediment deposition? If so, then I suggest rephrasing this sentence, since it makes it sound like subsidence rates can be no faster on 0.5 mm/yr, but subsidence rates depend on deposition rates, and thus could be faster in places with faster deposition.

Page 2, lines 17-19: It's not clear what is meant by the 40% in this sentence. I suggest clarifying this.

Page 2, line 25: I suggest changing "potential field" to "gravitational potential field", to be clear.

Equations 1 and 2: Technical point: The sea-level model computes changes in sea level due to changes in load, rather than the magnitude of sea level itself (see Equations 10 and 17 in Dalca et al., 2013). That is, in that notation, it computes Delta SL rather than SL, and it does so from Delta L rather than L. I suggest modifying Equations 1 and 2 in the the Methods section to clarify this.

Page 6, Figure 2: I suggest increasing the font size. The labels are too small to read

SED
easily in this map.

Page 7, lines 12-14: I suggest specifying how the eroded material was spatially distributed in the model. If it were proportional to ice sliding speed, then the eroded thickness would depend on spatial variations in ice sliding speed, which would require an ice flow model. Was that done? If so, how? Was it assumed that erosion under the ice sheet was spatially constant? If so, I suggest specifying that.

Page 7, Line 12: Contrary to this statement, recent evidence suggests that basal erosion scales with glacier sliding velocity squared, not sliding velocity to the first power (Herman et al., 2015, Science, v. 350, p. 193-195).

Page 7, Figure 3: Please add latitude and longitudes and a colorbar that defines what the colors mean.

Page 8, Figure 4: It's hard to tell where this is and what the scale is. Please modify this figure to include latitude and longitude.

Page 8, Figure 5: It's unclear what the colors and numbers represent. I suggest adding latitude and longitude grids and a colorbar, and expanding the text in the figure caption to explain what the colors and numbers mean.

Page 9, line 17: What is the time at which there are measurements? Is it the maximum at any time over the last  $\sim$ 10 kyr? Or the average over that time? Or the present? I suggest clarifying this in the caption.

Page 9, line 18: I suggest changing "gravity rate" to "rate of change of gravitational acceleration" for clarity.

Page 9, lines 18-21: It would be useful to show these boxes in a map in one of the figures to help show where these sites are.

Page 10, Figure 6 caption: I suggest specifying exactly what time LGM is taken to be here, since the timing of LGM is not universally agreed upon elsewhere in the literature.
For clarity, I also suggest changing "locations of Relative Sea Level data used in Fig. 7" to "Numbered black dots show locations of Relative Sea Level data in Figure 7."

Page 10, Figure 7: In most panels it's impossible to see a blue line. I assume that's because the red line and blue line are so close to one another that they overlap almost perfectly. If that's true, I suggest stating that in the figure caption.

Page 12, line 12: This states that the effects of sediment redistribution on sea level are comparable to those produced by water loading. This requires a citation, since changes due to water loading weren't shown in this study.

Page 12, line 20: I suggest noting that several tenths of a mm/yr is not negligible relative to modern globally averaged rates of sea-level change.

| SED |  |
|-----|--|
|-----|--|

---

## Author Comment (AC1) · 16 Jun 2017

Note: author reply in between « and »

This paper presents a clear and very well presented discussion of the incorporation of sediment loading effects into calculations of sea level change and other observables related to GIA. I think this discussion is warranted and contributes to the accuracy with which GIA effects are calculated. I would suggest this paper be accepted with minor technical revisions which I identify below:

« We thank Dr. Purcell for the effort and for the helpful comments. In our reply below , please note that line numbers refer to the document with tracked changes which is attached to this reply. »

[Figure]

1) Page 1, Line 29: "shown known" should simply be "shown"

« done »

2) Page 2, Equation 1: This formula is mis-labelled and is incorrect. The quantity represented here is not sea-level it is water-depth. In this context I should be multiplied by since it is not total ice thickness that is important, only that portion of ice thickness that displaces water.

« We follow the definitions of Dalca et al. (2013). In there (p. 460) the sea-level is defined as the height of the sea-surface equipotential relative to the solid surface. In that case the solid surface includes the ocean bottom + sediments + the top of the ice sheet. That means the topography will be equal to the negative of the sea level, i.e. the height of the solid surface with respect to the sea-surface equipotential. To make this more clear we add "Defining the sea-level as the difference between the equipotential corresponding to sea-level and the solid surface" p3 l18. Ice that is floating is not considered, cf the check in equation 4. »

3) Page 3, Equation 3: With the amendments suggested above the grounding line becomes SLj > 0

« Given the definition that we follow the check is first to see if there is ocean in the absence of ice. Then the check is to see if the weight of the ice height is larger than the weight of sea-level that it replaces, which is the G-R-H which is SL+I. To make the statement hopefully clearer we added "sea level is positive in the absence of ice and" on p4 l5. »

4) Page 5, Line 3: "entire" should be deleted or replaced with "all of" or "the whole of"

« changed to "all of" »

5) Page 5, Line 7: "largest of such" should be "largest such"

« done »

6) Figure 3: The projection used for this figure makes the geography a little difficult to interpret.

« This figure is moved to appendix B and latitude and longitude are added »

7) Figure 4:It is very hard to make out the modern coastline which makes the figure difficult to interpret.

« This figure is removed »

8) Figures 2, 3, 4, 5 and 6 use different projections. It would, I think, be better to standardise.

« figure 3 and 5 now have the same projection and figure 4 is removed. New figures 4 and 6 have the same projection »

9) While there is a discussion of the uncertainty in the viscosity model and the observational record the uncertainty in the ice load seems to be the most significant element for this analysis. If GIA data are used to constrain the ice sheet and sediment changes are not considered then the change in ice thickness will be biased to compensate for the neglected sedimentary load. The implicit assumption that ice thickness is fixed and known is inaccurate.

« We think we did not suggest that the ice load is perfectly known, only that when the ice thickness is inferred from GIA observations such as uplift rate, the estimated ice thickness is biased by neglecting sediment transport and other model errors. Model errors are now also mentioned in the conclusions p14 l6. »

Please also note the supplement to this comment:
http://www.solid-earth-discuss.net/se-2017-18/se-2017-18-AC1-supplement.pdf

**Supplement:**

**Effect of Ssediment loading in Fennoscandia and the Barents Sea during the last glacial cycleon GIA observations**

Wouter van der Wal1, Thijs IJpelaar2

1Faculty of Aerospace Engineering, Delft University of Technology, Delft, 2613 DH, Netherlands 2's Hertogenbosch, 5211 TL, Netherlands

Correspondence to: Wouter van der Wal (w.vanderwal@tudelft.nl)

**Abstract**

5

- Models for postglacial rebound, or Glacial Isostatic Adjustment (GIA) routinely include the effects of meltwater redistribution and changes in topography and coastlines. Since the sediment transport related to the dynamics of ice sheets may be comparable to that of sea level rise in terms of surface pressure, the loading effect of sediment deposition could cause measurable ongoing viscous readjustment. Here we study the loading effect of glacial induced sediment redistribution (GISR) related to the Weichselian ice sheet in Fennoscandia and the Barents Sea. The surface loading effect and its effect on the gravitational potential is modelled by including changes in sediment thickness in the sea level equation following the
- method of Dalca et al. (2013). Sediment displacement estimates are estimated in two different ways: (i) from a compilation of studies on smaller-local features: through mouth fans, large scale failure and basin flux and (ii) from output of a coupled ice-sediment model. To account for uncertainty in Earth's rheology three viscosity profiles are used.

It is found that sediment transport can lead to changes in relative sea level of at most several up to 2 meters in the last 6000 years with and the largerst effects occurring earlier in the deglaciation. This magnitude is below the error level of most of the

- 20 relative sea level data because those data are sparse and errors are larger for older dataincrease with length of time before present. The maximum-effect on present-day uplift rates is-reaches a few tenths of mm yr-1 in large parts of Norway and Sweden, which is around the measurement error of long-term GPS-GNSS (Global Navigation Satellite System) monitoring networks. The maximum effect on present-day gravity rates as measured by the GRACE (Gravity Recovery and Climate Experiment) satellite mission is up to tenths of  $\mu$ Gal yr-1, which is larger than the measurement error but below other error
- 25 sources. Since GISR causes systematic uplift in most mainland Scandinavia, including GISR in GIA models would improve interpretation of GPS-GNSS and GRACE observations there.

**1. Introduction**

Erosion in glaciated areas can beis larger than in non-glaciated regions (Hallet et al., 1996, Amantov et al. 2011 and references therein), and estimates for sediment deposition in glaciated regions vary from millimeters per year to centimeters per year close to glaciers (Elverhoi 1984; Finlayson 2012), which is comparable to global changes in relative sea level during

the last glacial cycle (Fairbanks 1989). Similarly to sea-level change, sedimentation rates are enhanced during deglaciation when run-off is larger (e.g. Tucker and Slingerland 1997, Ivins et al. 2007). These changes in surface loading can lead to changes in sea level and the Earth's solid surface during thousands of years because of visco-elastic relaxation driven by the mantle viscosity. This raises the question whether erosion and sedimentation that is enhanced during deglaciation affects

- 5 present-day GIA measurements. The loading effects of meltwater redistribution are routinely included in models of Glacial Isostatic Adjustment (GIA), but the loading effect of sediment transport is not. Of course, total sea level change is a global effect while sediment transport is a more local effect and displaced meltwater volume is much larger than the displaced sediment. On the other hand, sediment density is higher than water density, and effects of sediment transport during the last glacial cycle could influence present-day GIA measurements.
- 10 Several studies investigated the viscous response due to variation in past sedimentation rates. Ivins et al (2007) force their surface loading model with an estimate of postglacial sedimentation rates of 10 mm/year, compared to a background sedimentation rate over a glacial cycle of 1 mm/year. Their modelling predicted present-day subsidence of 1-8 mm/year although a more recent estimate reduces that amount to 0.5 mm yr-1 (Wolstencroft et al., 2014). Viscoelastic relaxation due to sediment deposition in the Indus River basin and Arabian Sea has been shown known to cause changes in relative sea-
- 15 level of up to 2 meters over 4000 years present day subsidence rates in the Arabian Sea and the Indus River basin (Ferrier et al., 20145). The effect is larger when the entire glacial cycle is considered, which is relevant when sea level data near the deltas are used to constrain global melt water volume (Ferrier et al. 2015). , and also the Gulf of Mexico (Ivins et al., 2007; Simms et al., 2013) although a recent estimate reduces that amount to 0.5 mm yr-1 (Wolstencroft et al., 2014).
- 20 While the aforementioned studies focused on sediment loading near river deltas far away from glaciated areas. With the large amount of sediment transport involved in glacier growth and melt it is possible that some of thecould also induce palaeo sea level changes and-present-day <del>observed</del> vertical motion near previously glaciated areas. We refer to material displaced by glacier growth and melt -is caused by past as glacial induced sediment redistribution (GISR)-rather than changes in ice or water load in the last glacial cycle. The When present-day observations are used to invert for infer\_viscosity or for-ice
- thickness in GIA models which means those inferences could be biased when GISR is not taken into account. The objective of this paper is to find out what is the effect of GISR during the Weichselian on GIA observables in Fennoscandia including the Barents Sea. The interest in this region stems from the fact that glacigenic sediment transport is large there (Elverhøi, 1984). The last 2,5 million years of glacial erosion resulted in a sediment layer of several kilometre thickness (Riis and Fjeldskaar 1992; Dowdeswell et al., 1996) with the last glaciation depositing sediment layers up to hundreds of meters
- 30 thickness (Elverhoi 1984). and s. Moreover, several observations of sediment deposition are available (e.g. Dowdeswell et al., 1996) from which the loading can be quantified (e.g. Dowdeswell 1996; Taylor et al. 2002). Here, the focus is on present-day uplift and, gravity rate of changes, and relative palaeo sea level data, which are routinely used to constrain GIA models.

Models exist which couple\_compute the sediment displacement as a result of to the movement of glaciers (e.g. Boulton 1996; de Winter et al., 2012), but since the ice sheet thickness, as in most GIA models, is not a dynamic model component, erosion is not coupled to the changes in ice thickness in this study. Instead, the amount of GISR is derived from literature on observed sediment deposits and reported output from a coupled ice-sediment model. Sediment being deposited in the ocean

- 5 will not only induce vertical motion, but also displace water and affect the gravity field. To model this effect we use the methodology of Dalca et al., (2013) to include sediment redistribution in the sea-level equation in a self-consistent way. Dalca et al., (2013) show that ignoring the the relative effect of time-varying ocean load resulting from sediment redistribution can result in errors in relative sea level (RSL) be of up to 40% in terms of relative sea level. The method will be discussed briefly in Sect. 2. After that, it is explained how different estimates of GISR are created. Next, sea level change.
- 10

25

drawn about the relevance of GISR in explaining GIA observations.

**2. Method**

The loading effect of ice and meltwater are routinely included in GIA models. The so-called sea-level equation is solved, which computes the sea-level distribution that accompanies a change in ice volume and corresponding changes in the Earth's

and uplift-deformation rates and gravity rates are calculated for the different sediment transport scenarios and conclusions are

shape and gravitational potential field (Farrell and Clark 1976; Mitrovica and Peltier 1991). The effect of sediments can also be included in the sea-level equation, as shown by Dalca et al., (2013). Here we follow Dalca et al., (2013), Kendall et al., (2005) and references therein. Only the key elements will be repeated here and some small differences will be pointed out.
 Defining the sea level as the difference between the equipotential corresponding to sea-level and the solid surface, Tthe sea level equation (SL) is written given by as

$$\quad SL = G - (R + H + I),$$

where G is the height of the equipotential surface coinciding with the sea level, R is the height of the Earth's crust, H is the thickness of sediments, I is the thickness of ice masses supported by land. The aim is to compute the changes in sea level

(1)

(2)

$$\Delta SL = \Delta G - \left(\Delta R - \Delta H - \Delta I\right)$$

as a result of a changes in The-total surface mass load *L*, which is defined as the sum of the changes in mass of water, ice and sediment:

$$\Delta L = \rho_w \Delta S + \rho_I \Delta I + \rho_H \Delta H , \qquad (32)$$

where  $\rho_w$ ,  $\rho_I$ ,  $\rho_H$  are the respective densities and S is the ocean thickness. Computing the change in sea level - $\Delta SL$  requires the change in equipotential surface and the solid Earth displacement which themselves depend on the change in sea level. The solution requires solving an integral equation which is usually done with an iterative approach. To solve the sea

level equation (2), loading changes are discretized at time steps of typically 1000 years. Two aspects need to be included to assure accurate representation of surface loads.

First, a check is performed Aat each time step *j* <del>a check is performed</del> to see whether ice is grounded or not by requiring that the ice starts to float when the pressure exerted by the ice (prescribed by the ice model) is equal to the pressure of the current sea level. Thus, floating occurs when sea level is positive in the absence of ice and

$$I_{j} < \left(SL_{j} + I_{j}\right) \frac{\rho_{w}}{\rho_{I}}.$$
(43)

Second, ocean-continent margins change with time to account for ice sheets replacing sea and vice versa, as well as the change in coastline as sea level rises next to a sloped coastline (see Kendall et el. 2005 and references therein). The change in coastline depends on the topography, which depends on the sea level change. This requires an iteration over the complete glacial cycle (the 'outer' iteration) on top of the Solving the sea level equation requires iteration to obtain the sea level

10

20

5

change for each time step (the 'inner' iteration, denoted with index i). To start the outer iteration over the glacial cycle, the pre-glacial topography  $T_0$  is assumed to be equal to the present-day topography  $T_p$ . With this topography, sea level at each time step is computed. To start the inner iteration for each time step,

an The initial guess for the change in ocean height is given by

15
$$\delta S_{j}^{i=0} = \delta h_{j}C_{j} - T_{p}(C_{j} - C_{j-1}) - \delta H_{j}$$
 (54)

where  $h_j$  is the uniform change in ocean height given by mass conservation with the current ocean basin and  $C_j$  is the ocean function at time  $t_j$ .  $\delta$  denotes a change in one time step different from the total change denoted by  $\Delta$ . Note that the change in sediment thickness is subtracted here because it is included in the definition of sea level.

A change in topography affects the location of coastlines and hence the distribution of melt water. At first the pre glacial topography  $T_0$  is assumed to be equal to the present day topography  $T_p$ . After computing sea level increments at all time steps, the topography is estimate can be improved using the total sea level rise:

$$T_0 = T_p + \Delta S L_p \tag{65}$$

With the improved pre-glacial topography, the computation of sea levels can be repeated (the -outer-iteration) until the preglacial topography reaches convergence. Erosion will also change the topography, but this effect is not included; the effect of

25 erosion on the location of coastlines is expected to be smaller than the loading changes of erosion and sediment deposition itself which are the main interest of this study.

To compute the change in equipotential surface and solid surface displacement the Earth's mechanical properties need to be known. Here we assume the Earth is radially symmetric, incompressible, and deforming according to a Maxwell rheology. For such an earth model, response functions for an impulse load can be computed in the spherical harmonic domain (Peltier

30 1974). An efficient solution method presented by Mitrovica and Peltier (1991) solves the sea level (2) in the spatial domain

while computing the response of the solid Earth in the spectral domain. This method requires transformations from the spatial to the spectral domain where some information is lost.

The effect of sediment redistribution is implemented in the numerical codes for the sea-level equation developed by Schotman (2007). A partial benchmark against other numerical solutions of the sea-level equations was carried out in Spada

5 et al. (2012). Rotational feedback is also included in the sea level equation following Wu and Peltier (1984) and Milne et al. (1998). The response of the Earth to surface loading for a radially symmetric Earth is computed with the multi-layer matrix propagation normal mode method (Vermeersen and Sabadini, 1997) which is benchmarked in Spada et al. (2011).

**3. Model inputs**

10 The computation requires several inputs, such as elastic parameters, the and viscosity of the Earth, and the ice and sediment distributions, which are discussed in the following subsections. For the present day topography ETOPO5 is used. Note that the computations of the sea level equation takes place in the spherical harmonic domain. The maximum spherical harmonic degree of 256 and size of the grid of quantities that are provided in the spatial domain, such as topography and surface load, is 256x512.

15

**3.1 Model inputs: viscosity and ice loading**

below the lithosphere, from 60 to 100 km depth.

In this study we consider a laterally homogeneous Earth model and vary the radial viscosity profile. As a reference model profile we use VM5a (Peltier and Drummond, 2008) which is an iteration of the VM2 profile model that is used in the creation of ICE-5G (Peltier 2004). As alternative models-profiles we select models-profiles that have been shown by Root et 20 al (2015b) to provide a good fit to sea level data, GPS and GRACE data in Fennoscandia-in-Root et al. (2015b). That study found two model sets viscosity profiles, one with higher viscosities in upper and lower mantle viscosity and one with lower viscosities. The fact that sediment loading is not taken into account to obtain viscosity profiles in Root et al. (2015b) will have a minor effect given that three very different viscosity profiles are selected to account for uncertainty in viscosity. Out of those sets we select the two models M8-128-150 and M4-16-80, where the first number denotes the upper mantle viscosity in  $10^{20}$  Pa s, the second number denotes the lower mantle viscosity in  $10^{20}$  Pa s and the third number denotes the lithosphere thickness in km. The three viscosity profiles are shown in Fig. 1. Note the lower viscosity  $10^{22}$  Pa s in VM5a just

Figure 1: The three viscosity profiles used in this study. M04 refers to viscosity profile M4-16-80, M08 refers to M8-128-150.

Since we are only interested in the effect of GISR the exact ice loading history is of less importance, and only influences the effect of GISR through the distribution of meltwater possibly replaced by sediment, which is a smaller effect than the sediment loading itself. For ice loading history the ICE-5G modelv1.2 (Peltier 2004) is selected which is provided with - Ttime steps are of 2,000 years from 120 kyr Before Present (B.P.) to 32 kyr B.P., 1,000 years from 32 to 17 kyr B.P. and 500 years from 17 kyr B.P. to present.

**3.2 Model inputs: sediment distribution**

5

In order to model the loading effect of GISR it needs to be known how much sediment is transported, where it came from

and where it is deposited. Erosion and deposit estimates for entire all of Scandinavia during the last glacial cycle are not readily available. Therefore, we created a map of sediment deposition from estimates in the literature of sediment volumes transported in smaller-local features: through mouth fans (TMFs) (i), large scale failures (ii) and basin flux (iii). Each of the features will be briefly discussed in the following.

(i) TMFs are places where rapid flowing ice streams at the end of the continental shelf converge, and where sediment is
 deposited off the shelf, see Figure 2 for the locations. Local observations come from sonar and seismic profiling
 (Dowdeswell et al., 1996) and coring (Saettem et al., 1992; Laberg and Vorren 1996; Taylor et al., 2002; Laberg et al., 2012). Other estimates come from modelling based on bathymetry, elevation and environmental conditions (Siegert and Dowdeswell 2002), but these are highly dependent on the amount of ice that is believed to have existed in the Barents and Kara Sea. Deposition also takes place on the shelf, but is probably smaller (Zieba et al. 2016)

The estimates are compiled in Table A1 in Appendix A.
 (ii) Large scale failures represent the sediment that is displaced after collapse of the slope. The largest of such events related to the Eurasian ice sheet is the Storegga slide. Haflidason et al. (2004; 2005) estimate the Holocene event to have a volume

of 2400-3200 km3 based on sonar scans and sedimentary cores. A compilation of the studies for this and other slides is provided in Table A2 in Appendix A.

---

## Author Comment (AC2) · 16 Jun 2017

author reply in between « and »

The manuscript is well written in terms of structure and English, but some improvements should be done. Many are of rather technical nature, but should be addressed carefully. I am willing to recommend publication after moderate revision.

« We are grateful to the reviewer for the effort and the helpful comments »

Title: as you deal with the Barents Sea, it should be included in the title. I'm reluctant to accept the term "sediment loading" in the title as you have erosional, thus unloading processes in your model as well. One can extent this discussion with the term "glacial-induced sediment redistribution" or "glacially induced sediment redistribution"

used later in the manuscript. Are you sure that your model includes glacially induced processes only? What about the "normal" erosion-transport-sedimentation process during the last glacial cycle up to now? In some areas in your research area was no or only little glacially influenced redistribution of rock material, in other areas the normal rock material redistribution is altered by the glaciation, but one may question what is "normal" in this sense. What is in common is certainly that you are talking about the main exogene processes. Nonetheless, I would like you to think about the term you are using and use it already in the title. My suggestion would be "Effects of exogene processes in Fennoscandia and the Barents Sea during the last glacial cycle on glacial isostatic adjustment observations".

« We agree that the title can be improved by a better specification of the region and processes involved. It is a good point that there is likely also 'normal sedimentation'. However, the word 'exogene' can mean different things and would not make the title clearer in our opinion. We opt for: "Effect of sediment redistribution in Fennoscandia and the Barents Sea during the last glacial cycle on GIA observations" »

Abstract: L8 remove "postglacial rebound, or"

« done »

L17 specify "at most several"

« replaced by "up to 2 m" which is the largest difference found at one of the RSL sites shown in figure 7. l 18 »

L20 change GPS -> GNSS and introduce abbreviation

« done »

L21 GRACE abbreviation unexplained

« done »

Main text: P2L20 uplift rates of deformation and gravity change

« changed to "deformation rates and gravity rates" »

Equation 1: what is SL?

« added before the equation »

Equation 2: what is L and S?

« L is added before equation 2 and S is now explained after the equation »

Equation 4: what is h_j and C_j?

« added »

P3 second last: comma missing: Earth, the ice

« added »

Figure 1: switch x and y axes, then the figure is easier to get

« done »

Section 3.2: although dealing with longer time spans, it'd be good to mention the works by Zieba and co-authors (Zieba et al. 2016, 2017).

« Thanks for pointing us to this reference. We think Zieba et al (2016) is relevant. a reference is added in p6l19 »

Figure 2: needs lat-lon information

« Figure 2 is used to show the labels of events. Figure B.1 and B.2 which show the shape of the events and the magnitude now have lat-lon information »

P6L15: add Sed1, 2 and 3 to the min, max and moderate explanation.

« done »

P7L2: Table A4

« done »

Figures 3, 4 and 5 are awkward due to missing lat-lon info, different projection and more (Figure 3: why is it location of erosion events in the caption, but spatial distribution of sediment contours when referred to in the text? What are the colors? Name of areas? Why is there no contribution from land?;

« Figure 4 is removed. Legend and lat-lon information is added to what is now figure B.1 and B.2. The word 'shape' is now used in the text and in the caption. A legend and latitude and longitude information are added. »

Figure 4: add information on blue and red, i.e. what is erosion what deposition. Amantov et al. (2011) seem to have a very fine resolution including major river systems, while Fig. 5 is rather coarse. Please explain in the text.

« Figure 4 is removed. In the caption of figure 5 (now B.2) is explained how values are obtained from spline interpolation »

Figure 5: color scale missing; figure not referenced in the text!)

« color scale is added, and a reference to the figure in Appendix B.2 »

As these figures are of rather technical nature, I strongly suggest to move this part to the supplement, i.e. P6L16 to P9L2, and summarize briefly in 1-2 sentences that you developed in addition to sed1,2,3 two more models based on Amantov et al., and that further info how you got these models can be found in the appendix. Also mention briefly the differences between Amantov 1 and 2.

« This is a good suggestion. We moved the material to Appendix B. »

To fill the space, I suggest to add a new Figure 3 showing a comparison of sed1,2,3 and Amantov1,2 similar as the old Figure 4, i.e. the total load change in metres of sediment thickness. Then you can explain this figure with the differences between the models.

P8L6: I understand that due to proprietary reasons Amantov's model is not available, however, it should be possible to clarify via e-mail to Amantov or Willy Fjeldskaar if the saturation is at 300 m or any value above...

« This has been clarified. 300 m was not a cut-off in the color scale, but was a maximum set for sediment thickness based on modelling. The plateau shape is not entirely realistic, this can be seen as reflecting the (considerable) uncertainty in sediment modelling (Amantov, personal communication). This is added to appendix B.2. Based on this information, the Amantov 2 model is no longer necessary and removed from the paper. The Amantov1 model is simply called 'Amantov'. »

P9L2: Please mark in Table A2, e.g. with a star, which features were added to Amantov 1 and 2.

« The features that were meant were the large scale failure events for Sed1 listed in table A.5. This has been clarified in Appendix B.2 »

P9L7: it is rather the Oslo Graben area than the center of the continent where the largest uplift is found.

« replaced by: "between south Norway and Sweden" »

Table 1: comparing Sed1,2,3 with Amantov 1,2 it appears that the first underestimate GISR by mainly about a factor of 2 in view of the latter

« added in the caption of figure 5 "It can be seen that the Amantov model results in values that are a factor two or three larger." »

P11L18: add "(not shown)"

« the figure has been added »

P12L4: the value of 0.016 might be ok from the math but is from my perspective overly optimistic in this regard. We know meanwhile from GNSS that the value is something like 10+ mm/a. Depending on longer time spans, reference frame issues, better correction models for the atmosphere, tropospheric delay etc. the value may change, but not to 9.5 mm/a or less or to way more than 11 mm/a. Anything in between is well covered by the uncertainties. Analysing GRACE instead, you'll see that the maximum value can be 0.9, 1.2 or even 1.5 microGal/a depending on the solution centre, used filter, time span etc. This is not at all covered by such a small value. You need to specify this for the reader. In view of the max gravity rate you get (Table 1) the sediments loading effect appears to be negligible for now. Also, even if you use a cut-off of degree 60, GRACE data are still filtered and thus you should filter the GISR result as well then.

« discussion added on page 11 »

P12L16: affects -> affected; remove "location"

« done »

Table A4: add min, max, moderate to Sed1,2,3.

« done »

Add data for Amantov1,2. What about the source areas? There is no information listed here nor in the main text. How are the continents treated? Please add!

« The source areas are taken from the reference, see figure B.2 and the comment on mass conservation in (now) appendix B.2 »

Finally: will the models (at least Sed1,2,3) be available for download somewhere so that other can use them?

« The sediment will be made available on the university webpage of the first author once the personal pages from our university are reinstated »

Please also note the supplement to this comment:
http://www.solid-earth-discuss.net/se-2017-18/se-2017-18-AC2-supplement.pdf

---

## Author Comment (AC3) · 16 Jun 2017

Please note: author reply in between « and »

The manuscript seeks to quantify the effect of post-LGM changes in sediment loading around Fennoscandia upon estimates of relative sea level, crustal motion and gravity changes. Sediment loading has only been addressed in a few papers, and this seems to be the first treatment for this region. The work is based on existing theory, although newly implemented within the authors' code. The conclusions are the effects of sediment loading are generally small, but may be important for present-day GPS and likely important for interpreting present-day gravity changes. The paper was written in a way that suggested it was put together in a rush. The major changes I request below are

[Figure]

largely related to figures and presentation. The English also needs some tightening up. I did not see any major flaws with the paper otherwise.

« We thank the reviewer for the effort and for the comments, which helped us a lot to improve the manuscript. »

Major remarks:

1. Figure 3 is not a scientific figure. it is a perspective image, but cannot be interpreted. it needs colours scale, graticule, length-scale, axis labels etc.

« The figure has been improved and moved to a new appendix B »

2. Figure 4 is not appropriate. the authors have digitised this, so replace with the two variants of the reconstruction shown properly.

« We removed the figure »

3. the authors use a sediment density of 2300 kg mЁЕ̧-3 but this choice is not defended or uncertainty tested. I presume sediment from different geological regions will have different densities.

« Amantov et al. (2011) provide water equivalent thickness and this is the load that is applied in the model. The statement about sediment density is removed because it is not relevant. Sediment density of 2300 kg/m3 is also used for the Sed1/2/3 models, this is explained in p 8 l 9 »

4. I believe there is work on pro-glacial lake loading (Fleming et al?) that is possibly relevant to mention.

« we searched but did not find relevant work from either the author or on pro-glacial lake loading that we thought was relevant to sediment loading »

5. the authors present two models with sediment flux and other models that focus on offshore load changes, but do not present a sum of the two - it wasn't clear why

« We think that the models are both equivalent in that they represent estimates of sedimentation and therefore they should not be added. The models Sed1/2/3 include local events as well as constant flux and conserves mass from source to sink areas (appendix B). The model of Amantov et al. (2011) is mass conserving but does not include large-scale failures which are added here (appendix B.2) »

P1 L15 "smaller features" - smaller than what? (also later in the paper)

« 'smaller' replaced by 'local' »

L19: "older data" - older than what? vague.

« rephrased to 'errors are larger for older data increase with time before present' »

"the maximum effect..." needs some geographical context.

« rephrased to 'reaches a few tenths of mm yr-1 in large parts of Norway and Sweden' »

P2 L10: relative sea level does not go with "present-day" earlier in the sentence and so this needs a rewrite

« done p2 bottom »

P3 Eq 4 - lower case delta is used instead of upper I think. h and C are not defined. S should be SL

« lower case is used for change over a time step. h and C are now defined, S was correct and is defined below equation 2. »

second last sentence : "parameters, the viscosity of the Earth, and the ice and ..."

« done »

final sentence: how are marine grounded ice sheets handled - this is worth explicit mention given recent controversies (Purcell et al)

« text added below equation 4. The description of Kendall et al. (2005) is followed »

P4 L2: "is degree 256 and the ..." - not clear what the grid is

« added p5 l13 »

L7/8: move "in Root ..." to "been shown by Root et al to. I note here these studies do not use sediment loading, so there's some issue here.

« done. Explanation added p5 l22 »

L18: not clear if the timesteps refer to the input ice loading or something to do with the computation

« clarified, p6 l6 »

P5: L3: "for the entirety of ..."

« changed to "all of' following another reviewer »

L12: "Large scale ..." - sentence begins without context.

« numbers are added to make clear that the parts refer to the itemization in the first paragraph »

This paragraph would benefit from the distance of the sediment transport being quantified. Figure 2 lacks any length-scale. it seems like a screenshot from Google Earth

« The source and sink areas that are used to create the sediment model are shown in the improved figure B.1 »

P6L6: the sediment loading is taken to be up to present-day. Could the authors clarify how this change in rate corresponds with completion of deglaciation? surely the change in sediment loading scaled down as the ice sheet decayed? Or is some sediment transported by melt-water of subglacier outburst floods?

« basin flux in the Sed1/2/3 models is enhanced before 10 ky before present, and has

a small value thereafter (p7l10). Sediment transport in the Amantov model is assumed here to follow ice mass change which means it is zero when deglaciation is complete in the region (appendix B.2)

L14: are -> were

« done »

L16: add comma after "sediment"; remove next comma.

« sentence rewritten, now on p23 l2 »

There are no contours on Fig 3.

« values within the shape are constant. A legend is added to what is now figure B.1 »

L18: what is the resolution of the grid?

« added 'in the 256x512 spatial grid' p23 l4 »

P7L7 find -> digitise.

« done »

L8: reference FIg 5 here.

« done »

L11: geoid shift - unclear what this means. over what period?

« added p23 l19 »

L13: 'the fact' - really a fact? is it linearly proportional or non-linearly? how may that affect sediment transport rates?

« the sentence is rephrased p 23 l 22 »

Figure 5: suggest this can move to supplementary material. suggest you make a proper map in QGIS! it needs a colour scale

« figure moved to appendix »

P9L2 results -> model input

« the sentence is removed »

L5: M4-160-80 is not a sediment model.

« replaced with 'viscosity profile' »

when is "displacement" relative to?

« 'sentence rephrased to agree with figure caption p9 l9 »

Table 1: sed1, 2,3 are not referred to in the text and need descriptions. the box described in the caption could be shown on a figure

« sed 1,2,3 now mentioned on p8 l7 box added to new figure 6 »

Figure 6 caption: clarify this is LGM to present loading changes

« done »

Figure 7: suggest a second set of y-axis to show the difference between the two curves
« done »

P11L8: could the authors give an example of where people have used GPS to infer ice load?

« rephrased and reference added p11 l14 »

Figure 8: the locations of the Lidberg GPS should be shown here or elsewhere

« locations are added to figure 5 »

P12L2: residuals of what?

« rephrased to' residuals obtained after fitting a trend, secular and annual signal to the monthly gravity fields' p12 l14 »

L6: around the same magnitude? ;how much larger?

« 'around' replaced by 'at the level of' »

L6-8: would be useful to understand typical % of signal

« numbers are inserted p 13 l 5 »

L13: effect is small on what?

« added 'on present-day GIA observables' p 13 l 16 »

L14:place over a large

« done »

L15: higher signal than what?

« added 'near areas of sediment deposition' p 13 l 19 »

L16: affects->affected; delete location

« done »

L18: suggest quantify the RSL effect here

« results added p13 l22 »

L22: not sure if these locations are all on figures

« Nova Zembla should be Svalbard, thanks for noticing »

L26: in -> from

« done »

L27: the *present-day* uplift

« done »

Please also note the supplement to this comment:
http://www.solid-earth-discuss.net/se-2017-18/se-2017-18-AC3-supplement.pdf

[Figure]

**Supplement:**

**Effect of sediment loading in Fennoscandia **and the Barents Sea** during the last glacial cycle **on GIA observations**

Wouter van der Wal[1], Thijs IJpelaar[2]

[1]Faculty of Aerospace Engineering, Delft University of Technology, Delft, 2613 DH, Netherlands
5   [2]'s Hertogenbosch, 5211 TL, Netherlands

*Correspondence to*: Wouter van der Wal (w.vanderwal@tudelft.nl)

**Abstract**

Models for  Glacial Isostatic Adjustment (GIA) routinely include the effects of meltwater
10   redistribution and changes in topography and coastlines. Since the sediment transport related to the dynamics of ice sheets may be comparable to that of sea level rise in terms of surface pressure, the loading effect of sediment deposition could cause measurable ongoing viscous readjustment. Here we study the loading effect of glacial induced sediment redistribution (GISR) related to the Weichselian ice sheet in Fennoscandia and the Barents Sea. The surface loading effect and its effect on the gravitational potential is modelled by including changes in sediment thickness in the sea level equation following the
15   method of Dalca et al. (2013). Sediment displacement estimates are estimated in two different ways: (i) from a compilation of studies on  local features: through mouth fans, large scale failure and basin flux and (ii) from output of a coupled ice-sediment model. To account for uncertainty in Earth's rheology three viscosity profiles are used.

It is found that sediment transport can lead to changes in relative sea level of up to 2 meters in the last 6000 years  and  largest effect occurring earlier in the deglaciation. This magnitude is below the error level of most of the
20   relative sea level data because those data are sparse and errors increase with length of time before present . The  effect on present-day uplift rates reaches a few tenths of mm yr$^{-1}$ in large parts of Norway and Sweden, which is around the measurement error of long-term  GNSS (Global Navigation Satellite System) monitoring networks. The maximum effect on present-day gravity rates as measured by the GRACE (Gravity Recovery and Climate Experiment) satellite mission is up to tenths of μGal yr$^{-1}$, which is larger than the measurement error but below other error
25   sources. Since GISR causes systematic uplift in most mainland Scandinavia, including GISR in GIA models would improve interpretation of  GNSS and GRACE observations there.

**1. Introduction**

Erosion in glaciated areas can be larger than in non-glaciated regions (Hallet et al., 1996, Amantov et al. 2011 and references therein), and estimates for sediment deposition in glaciated regions vary from millimeters per year to centimeters
30   per year close to glaciers (Elverhoi 1984; Finlayson 2012), which is comparable to global changes in relative sea level during

the last glacial cycle (Fairbanks 1989) . Similarly to sea-level change, sedimentation rates are enhanced during deglaciation when run-off is larger (e.g. Tucker and Slingerland 1997, Ivins et al. 2007). These changes in surface loading can lead to changes in sea level and the Earth's solid surface during thousands of years because of visco-elastic relaxation driven by the mantle viscosity. This raises the question whether erosion and sedimentation that is enhanced during deglaciation affects present-day GIA measurements. The loading effects of meltwater redistribution are routinely included in models of Glacial Isostatic Adjustment (GIA), but the loading effect of sediment transport is not. Of course, total sea level change is a global effect while sediment transport is a more local effect and displaced meltwater volume is much larger than the displaced sediment. On the other hand, sediment density is higher than water density, and effects of sediment transport during the last glacial cycle could influence present-day GIA measurements.

Several studies investigated the viscous response due to variation in past sedimentation rates. Ivins et al (2007) force their surface loading model with an estimate of postglacial sedimentation rates of 10 mm/year, compared to a background sedimentation rate over a glacial cycle of 1 mm/year. Their modelling predicted present-day subsidence of 1-8 mm/year although a more recent estimate reduces that amount to 0.5 mm yr$^{-1}$ (Wolstencroft et al., 2014). Viscoelastic relaxation due to sediment deposition in the Indus River basin and Arabian Sea has been shown known to cause changes in relative sea-level of up to 2 meters over 4000 years present-day subsidence rates in the Arabian Sea and the Indus River basin (Ferrier et al., 2014 5). The effect is larger when the entire glacial cycle is considered, which is relevant when sea level data near the deltas are used to constrain global melt water volume (Ferrier et al. 2015).

, and also the Gulf of Mexico (Ivins et al., 2007; Simms et al., 2013) although a recent estimate reduces that amount to 0.5 mm yr$^{-1}$ (Wolstencroft et al., 2014).

While the aforementioned studies focused on sediment loading near river deltas far away from glaciated areas, With the large amount of sediment transport involved in glacier growth and melt it is possible that some of the could also induce palaeo sea level changes and present-day observed vertical motion near previously glaciated areas. We refer to material displaced by glacier growth and melt is caused by past as glacial induced sediment redistribution (GISR) rather than changes in ice or water load in the last glacial cycle. The When present-day observations are used to invert for infer viscosity or for ice thickness in GIA models which means those inferences could be biased when GISR is not taken into account. The objective of this paper is to find out what is the effect of GISR during the Weichselian on GIA observables in Fennoscandia including the Barents Sea. The interest in this region stems from the fact that glacigenic sediment transport is large there (Elverhøi, 1984). The last 2,5 million years of glacial erosion resulted in a sediment layer of several kilometre thickness (Riis and Fjeldskaar 1992; Dowdeswell et al., 1996) with the last glaciation depositing sediment layers up to hundreds of meters thickness (Elverhoi 1984). and s. Moreover, several observations of sediment deposition are available (e.g. Dowdeswell et al., 1996) from which the loading can be quantified (e.g. Dowdeswell 1996; Taylor et al. 2002). Here, the focus is on present-day uplift and, gravity rate of changes, and relative palaeo sea level data, which are routinely used to constrain GIA models.

Models exist which  compute the sediment displacement as a result of  the movement of glaciers (e.g. Boulton 1996; de Winter et al., 2012), but since the ice sheet thickness, as in most GIA models, is not a dynamic model component, erosion is not coupled to the changes in ice thickness in this study. Instead, the amount of GISR is derived from literature on observed sediment deposits and reported output from a coupled ice-sediment model. Sediment being deposited in the ocean will not only induce vertical motion, but also displace water and affect the gravity field. To model this effect we use the methodology of Dalca et al., (2013) to include sediment redistribution in the sea-level equation in a self-consistent way. Dalca et al., (2013) show that ignoring the  time-varying ocean load resulting from sediment redistribution can result in errors in relative sea level (RSL)  of up to 40% . The method will be discussed briefly in Sect. 2. After that, it is explained how different estimates of GISR are created. Next, sea level change,  deformation rates and gravity rates are calculated for the different sediment transport scenarios and conclusions are drawn about the relevance of GISR in explaining GIA observations.

**2. Method**

The loading effect of ice and meltwater are routinely included in GIA models. The so-called sea-level equation is solved, which computes the sea-level distribution that accompanies a change in ice volume and corresponding changes in the Earth's shape and gravitational potential field (Farrell and Clark 1976; Mitrovica and Peltier 1991). The effect of sediments can also be included in the sea-level equation, as shown by Dalca et al., (2013). Here we follow Dalca et al., (2013), Kendall et al., (2005) and references therein. Only the key elements will be repeated here and some small differences will be pointed out.

Defining the sea level as the difference between the equipotential corresponding to sea-level and the solid surface, the sea level  (SL) is  given by

$$SL = G - (R + H + I),\tag{1}$$

where $G$ is the height of the equipotential surface coinciding with the sea level, $R$ is the height of the Earth's crust, $H$ is the thickness of sediments, $I$ is the thickness of ice masses supported by land. The aim is to compute the changes in sea level

$$\Delta SL = \Delta G - (\Delta R - \Delta H - \Delta I)\,)\tag{2}$$

as a result of a changes in  total surface mass load $L$, which is defined as the sum of the changes in mass of water, ice and sediment:

$$\Delta L = \rho_w \Delta S + \rho_I \Delta I + \rho_H \Delta H,\tag{3\;\sout{2}}$$

where $\rho_w$, $\rho_I$, $\rho_H$ are the respective densities and S is the ocean thickness. Computing the change in sea level $\Delta SL$ requires the change in equipotential surface and the solid Earth displacement which themselves depend on the change in sea level. The solution requires solving an integral equation which is usually done with an iterative approach. To solve the sea

level equation (2), loading changes are discretized at time steps of typically 1000 years. Two aspects need to be included to assure accurate representation of surface loads.

First, a check is performed at each time step $j$  to see whether ice is grounded or not by requiring that the ice starts to float when the pressure exerted by the ice (prescribed by the ice model) is equal to the pressure of the current sea level. Thus, floating occurs when sea level is positive in the absence of ice and

$$I_j < \left(SL_j + I_j\right)\frac{\rho_w}{\rho_I}. \tag{43}$$

Second, ocean-continent margins change with time to account for ice sheets replacing sea and vice versa, as well as the change in coastline as sea level rises next to a sloped coastline (see Kendall et el. 2005 and references therein). The change in coastline depends on the topography, which depends on the sea level change. This requires an iteration over the complete glacial cycle (the 'outer' iteration) on top of the  iteration to obtain the sea level change for each time step (the 'inner' iteration, denoted with index $i$).

To start the outer iteration over the glacial cycle, the pre-glacial topography $T_0$ is assumed to be equal to the present-day topography $T_p$. With this topography, sea level at each time step is computed. To start the inner iteration for each time step, an  initial guess for the change in ocean height is given by

$$\delta S_j^{i=0} = \delta h_j C_j - T_p \left(C_j - C_{j-1}\right) - \delta H_j \tag{54}$$

where $h_j$ is the uniform change in ocean height given by mass conservation with the current ocean basin and $C_j$ is the ocean function at time $t_j$. $\delta$ denotes a change in one time step different from the total change denoted by $\Delta$. Note that the change in sediment thickness is subtracted here because it is included in the definition of sea level.

 After computing sea level increments at all time steps, the topography estimate can be improved using the total sea level rise:

$$T_0 = T_p + \Delta SL_p \tag{65}$$

With the improved pre-glacial topography, the computation of sea levels can be repeated (the 'outer' iteration) until the pre-glacial topography reaches convergence. Erosion will also change the topography, but this effect is not included; the effect of erosion on the location of coastlines is  smaller than the loading changes of erosion and sediment deposition itself which are the main interest of this study.

To compute the change in equipotential surface and solid surface displacement the Earth's mechanical properties need to be known. Here we assume the Earth is radially symmetric, incompressible, and deforming according to a Maxwell rheology. For such an earth model, response functions for an impulse load can be computed in the spherical harmonic domain (Peltier 1974). An efficient solution method presented by Mitrovica and Peltier (1991) solves the sea level (2) in the spatial domain

while computing the response of the solid Earth in the spectral domain. This method requires transformations from the spatial to the spectral domain where some information is lost.

The effect of sediment redistribution is implemented in the numerical codes for the sea-level equation developed by Schotman (2007). A partial benchmark against other numerical solutions of the sea-level equations was carried out in Spada et al. (2012). Rotational feedback is also included in the sea level equation following Wu and Peltier (1984) and Milne et al. (1998). The response of the Earth to surface loading for a radially symmetric Earth is computed with the multi-layer matrix propagation normal mode method (Vermeersen and Sabadini, 1997) which is benchmarked in Spada et al. (2011).

**3. Model inputs**

The computation requires several inputs, such as elastic parameters, the and viscosity of the Earth, and the ice and sediment distributions, which are discussed in the following subsections. For the present day topography ETOPO5 is used. Note that the computations of the sea level equation takes place in the spherical harmonic domain. The maximum spherical harmonic degree of 256 and size of the grid of quantities that are provided in the spatial domain, such as topography and surface load, is 256x512.

**3.1 Model inputs: viscosity and ice loading**

In this study we consider a laterally homogeneous Earth model and vary the radial viscosity profile. As a reference model profile we use VM5a (Peltier and Drummond, 2008) which is an iteration of the VM2 profile model that is used in the creation of ICE-5G (Peltier 2004). As alternative models profiles we select models profiles that have been shown by Root et al (2015b) to provide a good fit to sea level data, GPS and GRACE data in Fennoscandia in Root et al. (2015b). That study found two model setsviscosity profiles, one with higher viscosities in upper and lower mantle viscosity and one with lower viscosities. The fact that sediment loading is not taken into account to obtain viscosity profiles in Root et al. (2015b) will have a minor effect given that three very different viscosity profiles are selected to account for uncertainty in viscosity. Out of those sets we select the two models M8-128-150 and M4-16-80, where the first number denotes the upper mantle viscosity in $10^{20}$ Pa s, the second number denotes the lower mantle viscosity in $10^{20}$ Pa s and the third number denotes the lithosphere thickness in km. The three viscosity profiles are shown in Fig. 1. Note the lower viscosity $10^{22}$ Pa s in VM5a just below the lithosphere, from 60 to 100 km depth.

[Figure]

**Figure 1: The three viscosity profiles used in this study. M04 refers to viscosity profile M4-16-80, M08 refers to M8-128-150.**

Since we are only interested in the effect of GISR the exact ice loading history is of less importance, and only influences the effect of GISR through the distribution of meltwater possibly replaced by sediment, which is a smaller effect than the sediment loading itself. For ice loading history the ICE-5G modelv1.2 (Peltier 2004) is selected which is provided with . Ttime steps are of 2,000 years from 120 kyr Before Present (B.P.) to 32 kyr B.P., 1,000 years from 32 to 17 kyr B.P. and 500 years from 17 kyr B.P. to present.

**3.2 Model inputs: sediment distribution**

In order to model the loading effect of GISR it needs to be known how much sediment is transported, where it came from and where it is deposited. Erosion and deposit estimates for entire all of Scandinavia during the last glacial cycle are not readily available. Therefore, we created a map of sediment deposition from estimates in the literature of sediment volumes transported in smaller local features: through mouth fans (TMFs) (i), large scale failures (ii) and basin flux (iii). Each of the features will be briefly discussed in the following.

(i) TMFs are places where rapid flowing ice streams at the end of the continental shelf converge, and where sediment is deposited off the shelf, see Figure 2 for the locations. Local observations come from sonar and seismic profiling (Dowdeswell et al., 1996) and coring (Saettem et al., 1992; Laberg and Vorren 1996; Taylor et al., 2002; Laberg et al., 2012). Other estimates come from modelling based on bathymetry, elevation and environmental conditions (Siegert and Dowdeswell 2002), but these are highly dependent on the amount of ice that is believed to have existed in the Barents and Kara Sea. Deposition also takes place on the shelf, but is probably smaller (Zieba et al. 2016)
The estimates are compiled in Table A1 in Appendix A.

(ii) Large scale failures represent the sediment that is displaced after collapse of the slope. The largest of such events related to the Eurasian ice sheet is the Storegga slide. Haflidason et al. (2004; 2005) estimate the Holocene event to have a volume

of 2400-3200 km$^3$ based on sonar scans and sedimentary cores. A compilation of the studies for this and other slides is provided in Table A2 in Appendix A.

[Figure]

**Figure 2: Schematic representation of the locations of Through Mouth Fans (purple), Large scale failures (brown) and basins (green).**

(iii) The TMFs and the large-scale failure are examples of local features. However, constant deposition of sediment from the source area to the basins (Figure 3) also takes place. During glaciations, sediment activity is increased, with thickness changes estimated to be 5 cm kyr$^{-1}$ and 2 cm kyr$^{-1}$ for the periods of 30-10 kyr B.P. and 10-0 kyr B.P. respectively (Taylor et al., 2002), which amounts to a total volume of 97 and 58 km$^3$ for the Norwegian basin and 485 and 290 km$^3$ for the Lofoten basin. There are also channels and canyon systems that are not captured by either the large scale individual features which

still provide larger sedimentation rates than the overall basin estimate. Estimates for the Lofoten channel system amount to 35 km$^3$ (Taylor et al., 2000), which is small enough that it can be neglected compared to the other events. The basin fluxes are given in Table A3 in Appendix A.

5    Conflicting estimates are stated for GISR volumes in Tables A1-A3 in Appendix A, for example the Byørnøya Through Mouth Fan in Table A1. Also the timing is uncertain for most events. Therefore, different sets of GISR loads were created, consisting of minimum, maximum and moderate estimates from the tables, which are labelled Sed1, Sed2 and Sed3, respectively. The time step is chosen to be the average of the time span given, rounded off to the nearest time step of the ice model. A value for sediment density of 2300 kg/m$^3$ is taken, in agreement with Amantov et al. (2008) and measurements for

10    shallow sediments (Zaborska et al. 2008). Uncertainty in the density is likely smaller than the uncertainty in thickness and timing which are represented here by the different sediment models.

[revised manuscript text omitted]

While the relative sea level over time also includes the geoid effect due to removed mass, the present-day uplift rate only represents the viscous readjustment due to past changes in surface loadings. The pattern of uplift rates is shown in Figure 8 for the Amantov and Sed1 GISR models. In Figure 8a uplift can be seen in the formerly glaciated region of Scandinavia and subsidence off-shore, corresponding to . Figure 8b mainly shows the effect of large scale failures. In both figures, the largest effects are off-shore where no GPS measurements can be made. To see the effect on observed uplift rates, the second data column in Table 1 shows the maximum effect of the sediment loading at any of the BIFROST sites of Lidberg et al. (2010). GISR is seen to always increase uplift rates, because most of the GPS measurement stations are in previously glaciated areas from where erosion took place. Thus, when GPS is used to draw conclusions on or validate glaciation history (e.g. Kierulf et al. 2014) the contribution of the ice may be overestimated. .

[Figure]

**Figure 5: Uplift rates caused by sediment redistribution according to GISR models Amantov1 (left) and Sed1 (right). Dots denote locations of GPS sites of Lidberg et al. (2010). Note the different color scales. Earth model M4-16-80 is used for both cases.**

Finally, the influence on gravity rates is also investigated (Figure 6) , as gravity rates derived from the GRACE satellite mission constrain GIA in Scandinavia (Steffen et al., 2008; van der Wal et al., 2011) and the Barents Sea (Root et al., 2015a). To compare with GRACE data a maximum spherical harmonic degree of 60 is used in the GIA model, which is the same truncation used in many GRACE studies. Similar to the uplift rate signal, Sed1 reflects the signal of the landslides offshore southern Norway and the Amantov‑ model has negative gravity rates west of the Barents Sea where sediment is deposited. To evaluate the magnitude of the effect, Table 1 provides the maximum gravity rate that occurs in the areas that are used for GIA studies, Scandinavia and the Barents Sea, see  Figure 6. The values can be judged by comparing against the GRACE measurements error. The measurement error of the gravity rates derived from GRACE is computed using the method of Wahr et al. (2004), assuming that residuals obtained after fitting a trend, secular and annual signal to the monthly gravity fields reflect noise. This method was shown to results in a similar error magnitude as calibrated standard deviations or a full variance-covariance matrix (van der Wal et al., 2010). The measurement error propagated to the trend has a value of 0.016 $\mu$Gal yr$^{-1}$ for a ten-year GRACE time series from January 2003 to July 2013. In Scandinavia sediment loading results in gravity rates  at the level of the measurement error for the Sed/2/3 models for the three arth models (between 0.002 and 0.014 $\mu$Gal yr$^{-1}$), and larger effects for the Amantov‑ model (between 0.007 and 0.027 $\mu$Gal yr$^{-1}$). However, gravity rates derived from GRACE are not only affected by measurement error. Using data from different processing centers, and using different correction models results in a larger spread in the gravity rate than the measurement error. Based on a comparison with GPS data the RMS error was estimated to be 1 mm/year in terms of uplift rate (van der Wal et al. 2011), which is around 10% of the maximum uplift rates. Assuming most of this reflects GRACE errors, this means that the gravity rate error is roughly 0.1 microGal/year. In this light, the values in Table 1 appear to be insignificant, but the effect of sediment loading is systematic.  Note that the direct attraction of the current rate of sedimentation in the Barents Sea is not included in our computations. The effect of  current sediment transport could become relevant for GRACE studies and should be investigated in future studies. .

[Figure]

**Figure 6: Gravity rate due to GISR according to the Amantov model for viscosity profile M4-16-80. Boxes in Scandinavia and the Barents Sea denote the areas for which the maximum gravity rate is given in Table 1.**

**5. Discussion and conclusions**

We investigated the effect of sediment transport during the past glaciation in Scandinavia on current GIA observables. The effect on present-day GIA observables is small compared to the effect of ice loads that are displaced during glaciation. Sediment uptake takes place over a large area, and deposition takes place in limited areas mostly confined to the ocean which can lead to locally higher signal near areas of sediment deposition. It was found that RSL data are not significantly affected by GISR because those data are located  near the shore, in between zones of erosion and deposition, and because a large part of the deposition takes place early in the deglaciation when the errors in RSL data are relatively large. At the LGM the effect is 18 m; at 6 kyear B.P. the effect is below 2 m, comparable to what was found for the Indus River basin (Ferrier et al. 2015).

Also, the effect on present-day uplift rate and gravity rates is limited; depending on the estimate for sediment transport that is used, the magnitude of GISR loading effects is near the measurement limit, several tenths of mm yr$^{-1}$ uplift rate and several tenths of μGal yr$^{-1}$ gravity rate. The magnitude is comparable to a recent estimate of subsidence in the Mississipi delta (Yu et al. 2012; Wolstencroft et al. 2014) but somewhat smaller because of the larger sediment deposition area for the Fennoscandian ice sheet, and the earlier demise of the ice sheet in the Barents Sea. The magnitude is smaller than a possible reference frame bias in GPS derived uplift rates, and in the presence of other GIA model uncertainties several tenths of mm/year does not appear to be significant.  Nevertheless, the effect is systematic, reducing uplift rates and gravity rates in the land areas of Fennoscandia and Svalbard and increasing gravity rates west of Fennoscandia and the Barents Sea. Thus, if uplift or gravity rates are used to infer viscosity profiles or ice thickness those estimates could be biased. Also, a few tenths of mm/year is not negligible compared to global average sea level change.

[revised manuscript text omitted]

**Appendix A: estimates of sediment displacement**

| Through Mouth Fan | Volume [km$^3$] | Timespan [ka B.P.] | Reference |
|---|---|---|---|
| Bellsund | -- | - | - |
| Byørnøya | 820-1100 | 22-17 | Taylor et al. (2002) |
| | 1360 | 21.5-17.5 | Laberg et al. (2012) |
| | 2700 | 30-8 | Siegert and Dowdeswell (2002) |
| | 4176 | Late Weichselian | Laberg and Vorren (1996) |
| | 4800 | 27-14 | Dowdeswell and Siegert (1999) |
| Franz Victoria | 500 | 30-8 | Siegert and Dowdeswell (2002) |
| Isfjorden | 22.5 | 30-0 | Elverhoi et al. (1995) |
| Kongsfjorden | - | - | - |
| North Sea | 800 | 30-0 | Taylor et al. (2002) |
| Storfjorden | 250 | 30-0 | Siegert and Dowdeswell (1999) |
| | 800 | 27-14 | Dowdeswell and Siegert (1999) |
| Svyataya Anna | Little | 30-8 | Siegert and Dowdeswell (1999) |
| | 2200 | 27-14 | Dowdeswell and Siegert (1999) |
| Voronin | little | 30-8 | Siegert and Dowdeswell (1999) |

Table A2: estimates for volume of sediment displaced in different events, timespan and reference. No estimates for the Bellsund and Kongsfjorden are available, but sediment transport there is expected to be relatively small.

| Large Scale Failure | Volume [km$^3$] | Timespan [ka B.P.] | Source |
|---|---|---|---|
| Andøya | 900 | 11-0 | Taylor et al. (2002) |
| Byørnøyenna | 1350 | 20-15 | Leynaud et al. (2009) |
| Hinlopen | 1200-1350 | 30 | Winkelmann et al. (2008) |
| North Faroes | 135-1700 | 9.85 | Taylor et al. (2002) |
| Storegga I | 3880 | 50-30 | Bugge et al. (1988) |
| Storegga II-III | 1700 | 8-6 | Bugge et al. (1988) |
| Storegga II-III | 2400-3200 | 7.25 | Haflidason et al. (2005) |
| Traenadjupet | 900-1900 | 4.2 | Taylor et al. (2002) |

Table A3: Estimates for volume of sediment displaced by large scale failures.

| | Rate [cm\ka] | Volume [km$^3$] | Timespan [ka B.P.] | Reference |
|---|---|---|---|---|
| Norwegian basin | 2 | 58 | 10 | Taylor et al. (2002) |
| Lofoten basin | 2 | 97 | 10 | Taylor et al. (2002) |
| Norwegian basin | 5 | 290 | 20 | Taylor et al. (2002) |
| Lofoten basin | 5 | 485 | 20 | Taylor et al. (2002) |
| Norwegian channel | - | 35 | 30-0 | |

**Table A4: estimates for volume of sediment displaced in the Lofoten and Norwegian basins and channel systems.**

| | Sed3 (max) | Sed1 (min) | Sed2 (moderate) | |
|---|---|---|---|---|
| **TMF** | Volume [km$^3$] | Volume [km$^3$] | Volume [km$^3$] | Time span [ka B.P.] |
| Byørnøya | 820 | 1360 | 4800 | 22-17 |
| North Sea | 800 | 800 | 800 | 30-0 |
| Storfjorden | 250 | 300 | 800 | 27-14 |
| Franz Victoria | 500 | 500 | 500 | 30-8 |
| Svyataya Anna | 100 | 250 | 2200 | 27-14 |
| **Large Scale Failure** | Volume [km$^3$] | Volume [km$^3$] | Volume [km$^3$] | Time span [ka BP] |
| Andøya | 900 | 900 | 900 | 8 |
| Bjørnøyrenna | 1350 | 1350 | 1350 | 18 |
| North Faroes | 135 | 1400 | 1700 | 10 |
| Storegga | 1700 | 2400 | 3200 | 7 |
| Traenadjupet | 900 | 1900 | 1900 | 4 |
| Hinlopen | 1200 | 1275 | 1350 | 30 |
| **Basin fluxes** | Volume [km$^3$] | Volume [km$^3$] | Volume [km$^3$] | Timespan [ka BP] |
| Norwegian basin (interglacial) | 58 | 58 | 58 | 9-0 |
| Lofoten basin (interglacial) | 97 | 97 | 97 | 9-0 |

| | | | | | |
|---|---|---|---|---|---|
| Norwegian basin (glacial) | | 290 | 290 | 290 | 30-10 |
| Lofoten basin (glacial) | | 485 | 485 | 485 | 30-10 |
| Channel | | 35 | 35 | 35 | 30-0 |

**Table A5: sediment estimates created from observations and estimates of individual estimates in tables A1 to A3, as described in the main text. Sed2 and 3 are the maximum and minimum models, respectively.**

**Appendix B.1: creation of spatial pattern from reported sediment displacement features**

To create the spatial distribution of erosion event shapes were drawn in the QGIS software package to match the figures in Taylor et al. (2002) and Winkelmann et al. (2008), see figure B.1. The shape files were converted to raster files with a sediment height for each point in the 256x512 spatial grid using MATLAB and GDAL. Across the source and sink areas of the sediments a uniform sediment height change is assumed so that the volume matches the estimates in Table A4. The areal extent is the largest source of uncertainty. Note that in reality deposition depends on distance from the source and also the area of deposition changes when the continental shelf is flooded or exposed (see e.g. Ferrier et al. 2015). In this study we opted to address the largest sources of uncertainty by using two independent estimates of GISR and three different Earth viscosity profiles.

**Appendix B.2: creation of spatial pattern from reported sediment displacement features**

The model of Amantov et al. (2011) couples ice sheet growth and erosion and is constrained by sedimentary and seismologic observations. The model accounts for enhanced erosion due to ice streams and erodability of various sub-ice surfaces among others. Sediment removal and deposition is shown in figure 3.6 of Amantov et al. (2011) in equivalent water thickness. Because part of the data is proprietary, the data for the time series and for creating the figure is not available. The software package QGIS was used to georeference the image and digitize the load at hundreds of points between which triangular interpolation was performed (figure B.2). The original model of Amantov et al. (2011) conserved mass, but in converting the graphics to a grid of sediment thickness values mass conservation was lost. We opted not to enforce mass conservation, as doing so would require further assumptions which introduce uncertainty. As a result there is a total geoid shift of around 13 cm over the glaciation, which causes a loading effect that is small compared to sediment thickness changes. To obtain a time series of sediment thickness it is assumed that the temporal variation of sediment transport follows that of total ice mass change, based on the common assumption that erosion rate is proportional to sliding velocity (see Herman et al., 2011). Recently, erosion has been found to be proportional to sliding velocity squared (Herman et al. 2015), which would enhance erosion rate during periods of largest ice change compared to our erosion history.

The color scale in 3.6 of Amantov et al. (2011) appears to be saturated at 300 m, but in fact 300 m is the best guess by Amantov et al. (2011) for maximum sediment deposition thickness and it set as maximum deposition in the modelling (Aleksey Amantov, personal communication). Finally, the large-scale failures for Sed1 from table A.5, which are not included in the model of Amantov et al. (2011), are added from Table A2.

[Figure]

**Figure B.1: shape of erosion events, derived from Taylor et al. (2002) and Winkelmann et al. (2008).**

[Figure]

**Figure B.2: Points captured in QGIS from Amantov et al. (2011) with value of sediment load expressed in meters of water load. Values of the points are interpolated using splines.**

---

## Author Comment (AC4) · 16 Jun 2017

« please note that author replies are in between « and ». Page numbers refer to the revised manuscript which is attached to this reply with track changes. We thank the reviewer for the comments which helped us to strengthen our manuscript, especially the introduction and conclusions. »

This study quantifies and presents the contribution of the sediments deposition in the GIA signal for the region of Fennoscandia for the first time. The authors implement a published methodology in their own codes and use sediment data retrieved from different (available) sources. Therefore the work is original and the results are sound. They clearly show that the sediment contribution for Fennoscandia is quite small and

comparable to measurements error, but it is a potential source of systematic bias.

« That is a good summary of our work »

I do not have major concerns; however the English and the overall presentation can be certainly improved. In particular I found the motivation in the introduction poor and not clear. Consequently also the discussion (and conclusion) sounds just drafted. For example it is not clear at all if the result was expected or not. Some sentences suggest that the expected effect should be more important than what actually found, but there is no discussion about it at all. The main motivation for this work is that the sediments deposition is expected to have a contribution comparable to the sea level feedback. However, on one hand the glacial erosion can build up kilometres of sediments in million years, so its absolute contribution is "large" (P 12, L 13 of the MS) but the deposition rate is actually small, usually few millimetres per year. On the other hand, the sea level feedback is a global effect that cannot be ignored, and the local sea level loading effect can be much larger than few millimetres per year. For example the retreating ice is replaced by the water, and if it is not correctly included the error is comparable with the ice loading effect. The difference between the effect of the sea level feedback and the sediment deposition should be addressed (in both introduction and discussion).

« This is a good comment. We improved the logic in the introduction, explaining how sedimentation compared to sea level and why it could be expected to affect GIA observables Please see the revised text in the introduction. »

The authors note that in some other part of the world the sediment deposition has been proven to cause present-day subsidence. The state of the art, of those studies in particular, should be described in the introduction and comparison should be made in the discussion. Has that deposition occurred with higher rate? Is it more localised? Is there any similarity in those studies to the Fennoscandia sediment deposition? Or what are the differences?

« A more extensive discussion of the magnitude of the viscous effect to sediment loading is provided in the introduction and comparisons are made in the discussion and conclusions, please refer to those sections. »

Points that need to be cleared or rephrased: P 1, L 25: "sediment deposition in glaciated regions vary from millimeters per year to centimeters per year, which is comparable to changes in sea level during the last glacial cycle". Do you mean the relative sea level? Do you mean local RSL in the same glaciated areas or in general the sea level effect in other regions?

« clarified and reference added in p2 l1 »

P 2, L 7: "2,5 million years resulted in a sediment layer of several km thickness". This is misleading. For example 5 km in 2.5 million years is 2 mm/yr.

« The number is replaced by an estimate for the last glaciation p2 l29 »

P 2, L 22: The Method section is not clear enough. Concepts defined in older studies are used here without properly recalling them. It is confusing even for people familiar with iterative method for solving the sea level equation.

« More explanation is added, please see the methods section »

In Eq. 1 the SL variable (supposedly Sea Level) is not defined.

« done »

P 3, L 2: In Eq. 2 the L is not defined. Add L: The total surface mass load "L" is defined as the sum...

« done »

P 3, L 5: "At each time step"... It is not declared that it is solved with a step-evolution. P 3, L 9: "inner iteration" is not clear since you have not defined the method as iterative in a general way. P 3, Eq. 4: $C_j$ is used without defining it.

« These are included in the revised explanation of the method »

P 3, L 18: Why "the effect of erosion on the location of coastlines is expected to be small"?

« rephrased to 'smaller than the loading changes of erosion and sediment deposition itself, which is the topic of this study' p4 l 25 »

P 3, L 24: "The response of the Earth : : :. is computed with the normal mode method (Vermeersen and Sabadini, 1997) which is benchmarked in Spada et al. (2011)." Which code? Most codes benchmarked in Spada et al. (2011) implement the normal mode. And is therefore the code an incompressible model?

« added 'multi-layer matrix propagation' on p5 l6 and 'incompressible' added p4 l28 »

P 4, L 17: ": : : and only influences the effect of GISR through the distribution of meltwater possibly replaced by sediment, which is a small effect." What has a small effect? The melt water replaced by sediments? The fact that is small is not self-evident.

« changed to 'smaller than sediment loading itself' p 6 l 5 »

P 7, L 5: Is it so difficult to get Amantov data? And which is the original data accuracy?

« yes. The authors of that study have been helpful in explaining their method, but the model output described in the paper is proprietary. Amantov noted that it is not possible to put an accuracy on the model output. Our efforts therefore focussed on obtained two independent sets of GISR estimates. »

P 8 Figure 4: I don't think Figure 4 is really necessary. You can easily indicate in Fig. 5 the areas where the colour scale from Amantov picture is saturated. So you spare a picture and the issue of copy rights.

« figure 4 is removed »

P 8 Figure 5: This picture needs a colour scale bar.

« added, this now figure B.2 »

P 10, Fig 7. RSL curves: the difference is not visible at all. The authors could make the red dashed. Since there's no visible difference this picture doesn't give more info than what you can tell in the text

« line is made dashed and the difference is also shown in what is now figure 4. »

P 11, L 8: "Thus, by interpreting the GPS rates as only resulting from ice unloading, the contribution from the ice is overestimated. This could result in biased inferences of ice thickness." This is one of the possible relevant effects and it should be (at least roughly) quantified. Is it a 1% overestimate or is it a 10%? However considering that most contribution of sediments to the uplift is in the sea and it is within the error, other source of error (such as the GPS reference frame error) could cancel it.

« We did not do the inversion for ice history, but numbers are given for the magnitude of the uplift rate in table 1. Discussion is added to 'discussion and conclusion' on the significance of the effect. »

P 11, L 19: "To evaluate the magnitude of the effect, Table 1 provides the maximum gravity rate that occurs in the areas that are used for GIA studies, Scandinavia and the Barents Sea" Why not showing this with a picture?

« figure 6 has been added. »

P 12, L 13: "Although the amount of sediment transported is LARGE, the effect is small compared to the ice loads that" Large compared to : : :? This sentence needs to be rephrased and this small effect should be explained (or discussed better). It's not clear if such small effect was expected or not. From the introduction I would guess that a larger effect was expected. So what is the main cause for a "large" amount of sediments to produce a "small effect"?

« 'large' is removed. Comparison with earlier studies is added further on in the discussion and conclusions »

Minor comments P 1, L 29: "has been shown known" -> shown or known?

« done »

P 2, L 7: "2,5" -> "2.5"

« sentence removed »

P 2, L 12: "Models exist which couple : : :" does not sound like good English

« rephrased »

P 2 , L 23: "The so-called sea-level equation is solved, which computes : : :" does not sound like good English

« the sentence does not appear to be problematic to us »

P 4, L 9 and Figure 1: "M8-128-150 and M4-16-80" are these M04 and M08 in the figure? The names in the legend are not fully self-explanatory.

« names added to the caption of figure 1 »

P 4, L 18: "modelv1.2" what is this? I understand that is a sort of update of the ICE-5G, is there a link or more detail to find that? What's the difference from the original?

« This is in fact the version of ICE-5G that was provided and widely used and referred to with Peltier (2004), but not always labelled correctly as v1.2. »

P 6, L 17: Why citing QGIS and MATLAB, GDAL, which are only tool that implement algorithm, and not instead citing the specific algorithms used?

« The contour drawing is done by hand. Converting shape to raster is a standard procedure for GIS software and not much value is added if the algorithms internal to the software are stated. »

P 9 Table 1: This table is not immediate to read. nn/nn/nn is visually not effective. Spaces could help, for example nn / nn / nn or even using 3 sub columns for each main

column

« / is replaced by | with extra spaces. »

P 12, L 9: "Sediment transport from the Barents Sea to the west would have the opposite effect on the gravity rate but as of yet sediment transport is not yet detected in GRACE measurements." This sentence must be rephrased, I had to read it three times to understand it.

« rephrased p12 l18 »

P 12, L 13: "ice loads" -> "ice loads effect"..

« rephrased »

P 12, L16: "RSL data are not significantly affects" -> "RSL data are not significantly affected" and "because those data are located location near..." -> "because those data are located near..."

« done »

Please also note the supplement to this comment:
http://www.solid-earth-discuss.net/se-2017-18/se-2017-18-AC4-supplement.pdf

---

## Author Comment (AC5) · 16 Jun 2017

The main purpose of this study is to quantify sea-level responses to sediment redistribution caused by ice sheets in Fennoscandia over the last glacial cycle. To do so, the authors apply a recent sea-level model (Dalca et al., 2013), which computes sealevel responses to sediment erosion and deposition. The main finding is contained in Figure 7, which shows that sea-level responses to sediment redistribution are small in this region, such that accounting for sediment redistribution does not significantly help resolve differences between observed and modeled relative sea-level histories. This is a useful finding and the main strength of this study.

The manuscript has several weaknesses that I suggest the authors address before

publication, most of which have to do with the presentation of the material. As I describe below, a number of items in the text are unclear, and most of the figures require major modification before they can be understood, particularly Figures 3-5. I do not have major scientific concerns about the study, but two minor concerns are that the study did not conserve sediment mass, and it's not clear how eroded material was spatially distributed, which would make it difficult to reproduce the results of this study. I suggest the authors expand on these points in the text. Overall, I suggest that this study will be of interest to a number of readers in Solid Earth after major revision.

« We thank the reviewer for the thoughtful review. We have improved the presentation following your comments and those of the other reviewers. The issue of mass conservation was, we think, sufficiently addressed in the manuscript (now in appendix B.2) and the other reviewers seem to agree. Indeed the sediment models would be difficult to reproduce as there is some 'hand picking' to derive contours, because of unavailability of source data. It would be hard to describe the redrawing procedure. We will make our sediment distribution available on our institution website such that they can be used or checked. A brief discussion on the issue of spatial pattern of deposition and erosion is added in Appendix B.1 »

Additional comments Page 1, line 17: I suggest specifying the timescale over which changes in relative sea level can be as large as several meters. Is this the integrated sea-level change from the Last Interglacial to the present?

« added 'in the last 6000 years' p1 l18 »

Page 1, line 25: I suggest rephrasing this sentence, since glacial erosion is not always faster than non-glacial erosion. Glaciers frozen to their beds, for example, can inhibit erosion, rather than accelerating erosion.

« changed to 'can be' p1l28. The relation to run-off is mentioned in the next sentence. »

Page 2, line 1: Does "that amount" in this sentence refer to subsidence rates due to sediment deposition? If so, then I suggest rephrasing this sentence, since it makes it sound like subsidence rates can be no faster on 0.5 mm/yr, but subsidence rates depend on deposition rates, and thus could be faster in places with faster deposition.

« text added in p2 l10-13 which hopefully makes clear that the viscous effect is referred to »

Page 2, lines 17-19: It's not clear what is meant by the 40% in this sentence. I suggest clarifying this.

« rephrased p3 l8 »

Page 2, line 25: I suggest changing "potential field" to "gravitational potential field", to be clear.

« done »

Equations 1 and 2: Technical point: The sea-level model computes changes in sea level due to changes in load, rather than the magnitude of sea level itself (see Equations 10 and 17 in Dalca et al., 2013). That is, in that notation, it computes Delta SL rather than SL, and it does so from Delta L rather than L. I suggest modifying Equations 1 and 2 in the the Methods section to clarify this.

« Thank you for pointing that out. The method section is improved, also in response to the other reviewers »

Page 6, Figure 2: I suggest increasing the font size. The labels are too small to read easily in this map.

« We have enlarged the figure somewhat and will reproduce the figure with better quality if necessary. »

Page 7, lines 12-14: I suggest specifying how the eroded material was spatially distributed in the model. If it were proportional to ice sliding speed, then the eroded

thickness would depend on spatial variations in ice sliding speed, which would require an ice flow model. Was that done? If so, how? Was it assumed that erosion under the ice sheet was spatially constant? If so, I suggest specifying that.

« What is referred to in the text as the Amantov model is output of the model of Amantov et al. (2011) which includes an ice flow model. Some more details on the Amantov model are now given in appendix B.2. »

Page 7, Line 12: Contrary to this statement, recent evidence suggests that basal erosion scales with glacier sliding velocity squared, not sliding velocity to the first power (Herman et al., 2015, Science, v. 350, p. 193-195).

« The reference has been added p23 l23. We used the proportionality between erosion and sliding velocity loosely to obtain a time series of deposition. The relation between ice volume change and sliding velocity is also unknown. »

Page 7, Figure 3: Please add latitude and longitudes and a colorbar that defines what the colors mean.

« done, please see new figure B.1 »

Page 8, Figure 4: It's hard to tell where this is and what the scale is. Please modify this figure to include latitude and longitude.

« The figure is removed »

Page 8, Figure 5: It's unclear what the colors and numbers represent. I suggest adding latitude and longitude grids and a colorbar, and expanding the text in the figure caption to explain what the colors and numbers mean.

« done, please see new figure B.2 »

Page 9, line 17: What is the time at which there are measurements? Is it the maximum at any time over the last _10 kyr? Or the average over that time? Or the present? I suggest clarifying this in the caption.

« reference to figure 5 is added in the caption of table 1 »

Page 9, line 18: I suggest changing "gravity rate" to "rate of change of gravitational acceleration" for clarity.

« done »

Page 9, lines 18-21: It would be useful to show these boxes in a map in one of the figures to help show where these sites are.

« figure 7 has been added »

Page 10, Figure 6 caption: I suggest specifying exactly what time LGM is taken to be here, since the timing of LGM is not universally agreed upon elsewhere in the literature.

« added '(26,000 years B.P. in the ICE-5G model)' to the caption of figure 6 »

For clarity, I also suggest changing "locations of Relative Sea Level data used in Fig. 7" to "Numbered black dots show locations of Relative Sea Level data in Figure 7."

« added to what is now figure 4 »

Page 10, Figure 7: In most panels it's impossible to see a blue line. I assume that's because the red line and blue line are so close to one another that they overlap almost perfectly. If that's true, I suggest stating that in the figure caption.

« A dashed line and a line with the difference is added »

Page 12, line 12: This states that the effects of sediment redistribution on sea level are comparable to those produced by water loading. This requires a citation, since changes due to water loading weren't shown in this study.

« the statement is removed »

Page 12, line 20: I suggest noting that several tenths of a mm/yr is not negligible relative to modern globally averaged rates of sea-level change.

« added p 14 l 10 »

Please also note the supplement to this comment:
http://www.solid-earth-discuss.net/se-2017-18/se-2017-18-AC5-supplement.pdf

**Supplement:**

**Effect of sediment loading in Fennoscandia and the Barents Sea during the last glacial cycle on GIA observations**

Wouter van der Wal[1], Thijs IJpelaar[2]

[1]Faculty of Aerospace Engineering, Delft University of Technology, Delft, 2613 DH, Netherlands
[2]'s Hertogenbosch, 5211 TL, Netherlands

*Correspondence to*: Wouter van der Wal (w.vanderwal@tudelft.nl)

**Abstract**

Models for  Glacial Isostatic Adjustment (GIA) routinely include the effects of meltwater redistribution and changes in topography and coastlines. Since the sediment transport related to the dynamics of ice sheets may be comparable to that of sea level rise in terms of surface pressure, the loading effect of sediment deposition could cause measurable ongoing viscous readjustment. Here we study the loading effect of glacial induced sediment redistribution (GISR) related to the Weichselian ice sheet in Fennoscandia and the Barents Sea. The surface loading effect and its effect on the gravitational potential is modelled by including changes in sediment thickness in the sea level equation following the method of Dalca et al. (2013). Sediment displacement estimates are estimated in two different ways: (i) from a compilation of studies on  local features: through mouth fans, large scale failure and basin flux and (ii) from output of a coupled ice-sediment model. To account for uncertainty in Earth's rheology three viscosity profiles are used.

It is found that sediment transport can lead to changes in relative sea level of up to 2 meters in the last 6000 years and largest effect occurring earlier in the deglaciation. This magnitude is below the error level of most of the relative sea level data because those data are sparse and errors increase with length of time before present . The  effect on present-day uplift rates reaches a few tenths of mm yr$^{-1}$ in large parts of Norway and Sweden, which is around the measurement error of long-term  GNSS (Global Navigation Satellite System) monitoring networks. The maximum effect on present-day gravity rates as measured by the GRACE (Gravity Recovery and Climate Experiment) satellite mission is up to tenths of μGal yr$^{-1}$, which is larger than the measurement error but below other error sources . Since GISR causes systematic uplift in most mainland Scandinavia, including GISR in GIA models would improve interpretation of  GNSS and GRACE observations there.

**1. Introduction**

Erosion in glaciated areas can be larger than in non-glaciated regions (Hallet et al., 1996, Amantov et al. 2011 and references therein), and estimates for sediment deposition in glaciated regions vary from millimeters per year to centimeters per year close to glaciers (Elverhoi 1984; Finlayson 2012), which is comparable to global changes in relative sea level during

the last glacial cycle (Fairbanks 1989) . Similarly to sea-level change, sedimentation rates are enhanced during deglaciation when run-off is larger (e.g. Tucker and Slingerland 1997, Ivins et al. 2007). These changes in surface loading can lead to changes in sea level and the Earth's solid surface during thousands of years because of visco-elastic relaxation driven by the mantle viscosity. This raises the question whether erosion and sedimentation that is enhanced during deglaciation affects present-day GIA measurements. The loading effects of meltwater redistribution are routinely included in models of Glacial Isostatic Adjustment (GIA), but the loading effect of sediment transport is not. Of course, total sea level change is a global effect while sediment transport is a more local effect and displaced meltwater volume is much larger than the displaced sediment. On the other hand, sediment density is higher than water density, and effects of sediment transport during the last glacial cycle could influence present-day GIA measurements.

Several studies investigated the viscous response due to variation in past sedimentation rates. Ivins et al (2007) force their surface loading model with an estimate of postglacial sedimentation rates of 10 mm/year, compared to a background sedimentation rate over a glacial cycle of 1 mm/year. Their modelling predicted present-day subsidence of 1-8 mm/year although a more recent estimate reduces that amount to 0.5 mm yr$^{-1}$ (Wolstencroft et al., 2014). Viscoelastic relaxation due to sediment deposition in the Indus River basin and Arabian Sea has been shown known to cause changes in relative sea-level of up to 2 meters over 4000 years present-day subsidence rates in the Arabian Sea and the Indus River basin (Ferrier et al., 201 4 5). The effect is larger when the entire glacial cycle is considered, which is relevant when sea level data near the deltas are used to constrain global melt water volume (Ferrier et al. 2015).
, and also the Gulf of Mexico (Ivins et al., 2007; Simms et al., 2013) although a recent estimate reduces that amount to 0.5 mm yr$^{-1}$ (Wolstencroft et al., 2014).

While the aforementioned studies focused on sediment loading near river deltas far away from glaciated areas, With the large amount of sediment transport involved in glacier growth and melt it is possible that some of the could also induce palaeo sea level changes and present-day observed vertical motion near previously glaciated areas. We refer to material displaced by glacier growth and melt is caused by past as glacial induced sediment redistribution (GISR) rather than changes in ice or water load in the last glacial cycle. The When present-day observations are used to invert for infer viscosity or for ice thickness in GIA models which means those inferences could be biased when GISR is not taken into account. The objective of this paper is to find out what is the effect of GISR during the Weichselian on GIA observables in Fennoscandia including the Barents Sea. The interest in this region stems from the fact that glacigenic sediment transport is large there (Elverhøi, 1984). The last 2,5 million years of glacial erosion resulted in a sediment layer of several kilometre thickness (Riis and Fjeldskaar 1992; Dowdeswell et al., 1996) with the last glaciation depositing sediment layers up to hundreds of meters thickness (Elverhoi 1984). and s. Moreover, several observations of sediment deposition are available (e.g. Dowdeswell et al., 1996) from which the loading can be quantified (e.g. Dowdeswell 1996; Taylor et al. 2002). Here, the focus is on present-day uplift and, gravity rate of changes, and relative palaeo sea level data, which are routinely used to constrain GIA models.

Models exist which  compute the sediment displacement as a result of  the movement of glaciers (e.g. Boulton 1996; de Winter et al., 2012), but since the ice sheet thickness, as in most GIA models, is not a dynamic model component, erosion is not coupled to the changes in ice thickness in this study. Instead, the amount of GISR is derived from literature on observed sediment deposits and reported output from a coupled ice-sediment model. Sediment being deposited in the ocean will not only induce vertical motion, but also displace water and affect the gravity field. To model this effect we use the methodology of Dalca et al., (2013) to include sediment redistribution in the sea-level equation in a self-consistent way. Dalca et al., (2013) show that ignoring the  time-varying ocean load resulting from sediment redistribution can result in errors in relative sea level (RSL)  of up to 40% . The method will be discussed briefly in Sect. 2. After that, it is explained how different estimates of GISR are created. Next, sea level change,  deformation rates and gravity rates are calculated for the different sediment transport scenarios and conclusions are drawn about the relevance of GISR in explaining GIA observations.

**2. Method**

The loading effect of ice and meltwater are routinely included in GIA models. The so-called sea-level equation is solved, which computes the sea-level distribution that accompanies a change in ice volume and corresponding changes in the Earth's shape and gravitational potential field (Farrell and Clark 1976; Mitrovica and Peltier 1991). The effect of sediments can also be included in the sea-level equation, as shown by Dalca et al., (2013). Here we follow Dalca et al., (2013), Kendall et al., (2005) and references therein. Only the key elements will be repeated here and some small differences will be pointed out.

Defining the sea level as the difference between the equipotential corresponding to sea-level and the solid surface, the sea level  (SL) is  given by

$$SL = G - (R + H + I),\tag{1}$$

where $G$ is the height of the equipotential surface coinciding with the sea level, $R$ is the height of the Earth's crust, $H$ is the thickness of sediments, $I$ is the thickness of ice masses supported by land. The aim is to compute the changes in sea level

$$\Delta SL = \Delta G - (\Delta R - \Delta H - \Delta I))\tag{2}$$

as a result of a changes in  total surface mass load $L$, which is defined as the sum of the changes in mass of water, ice and sediment:

$$\Delta L = \rho_w \Delta S + \rho_I \Delta I + \rho_H \Delta H,\tag{3}$$

where $\rho_w$, $\rho_I$, $\rho_H$ are the respective densities and S is the ocean thickness. Computing the change in sea level $\Delta SL$ requires the change in equipotential surface and the solid Earth displacement which themselves depend on the change in sea level. The solution requires solving an integral equation which is usually done with an iterative approach. To solve the sea

level equation (2), loading changes are discretized at time steps of typically 1000 years. Two aspects need to be included to assure accurate representation of surface loads.

First, a check is performed at each time step $j$  to see whether ice is grounded or not by requiring that the ice starts to float when the pressure exerted by the ice (prescribed by the ice model) is equal to the pressure of the current sea level. Thus, floating occurs when sea level is positive in the absence of ice and

$$I_j < \left(SL_j + I_j\right)\frac{\rho_w}{\rho_I}. \tag{4}$$

Second, ocean-continent margins change with time to account for ice sheets replacing sea and vice versa, as well as the change in coastline as sea level rises next to a sloped coastline (see Kendall et el. 2005 and references therein). The change in coastline depends on the topography, which depends on the sea level change. This requires an iteration over the complete glacial cycle (the 'outer' iteration) on top of the  iteration to obtain the sea level change for each time step (the 'inner' iteration, denoted with index $i$).

To start the outer iteration over the glacial cycle, the pre-glacial topography $T_0$ is assumed to be equal to the present-day topography $T_p$. With this topography, sea level at each time step is computed. To start the inner iteration for each time step, an  initial guess for the change in ocean height is given by

$$\delta S_j^{i=0} = \delta h_j C_j - T_p \left(C_j - C_{j-1}\right) - \delta H_j \tag{5}$$

where $h_j$ is the uniform change in ocean height given by mass conservation with the current ocean basin and $C_j$ is the ocean function at time $t_j$. $\delta$ denotes a change in one time step different from the total change denoted by $\Delta$. Note that the change in sediment thickness is subtracted here because it is included in the definition of sea level.

 After computing sea level increments at all time steps, the topography estimate can be improved using the total sea level rise:

$$T_0 = T_p + \Delta SL_p \tag{6}$$

With the improved pre-glacial topography, the computation of sea levels can be repeated (the 'outer' iteration) until the pre-glacial topography reaches convergence. Erosion will also change the topography, but this effect is not included; the effect of erosion on the location of coastlines is  smaller than the loading changes of erosion and sediment deposition itself which are the main interest of this study.

To compute the change in equipotential surface and solid surface displacement the Earth's mechanical properties need to be known. Here we assume the Earth is radially symmetric, incompressible, and deforming according to a Maxwell rheology. For such an earth model, response functions for an impulse load can be computed in the spherical harmonic domain (Peltier 1974). An efficient solution method presented by Mitrovica and Peltier (1991) solves the sea level (2) in the spatial domain

while computing the response of the solid Earth in the spectral domain. This method requires transformations from the spatial to the spectral domain where some information is lost.

The effect of sediment redistribution is implemented in the numerical codes for the sea-level equation developed by Schotman (2007). A partial benchmark against other numerical solutions of the sea-level equations was carried out in Spada et al. (2012). Rotational feedback is also included in the sea level equation following Wu and Peltier (1984) and Milne et al. (1998). The response of the Earth to surface loading for a radially symmetric Earth is computed with the multi-layer matrix propagation normal mode method (Vermeersen and Sabadini, 1997) which is benchmarked in Spada et al. (2011).

**3. Model inputs**

The computation requires several inputs, such as elastic parameters, the and viscosity of the Earth, and the ice and sediment distributions, which are discussed in the following subsections. For the present day topography ETOPO5 is used. Note that the computations of the sea level equation takes place in the spherical harmonic domain. The maximum spherical harmonic degree of 256 and size of the grid of quantities that are provided in the spatial domain, such as topography and surface load, is 256x512.

**3.1 Model inputs: viscosity and ice loading**

In this study we consider a laterally homogeneous Earth model and vary the radial viscosity profile. As a reference model profile we use VM5a (Peltier and Drummond, 2008) which is an iteration of the VM2 profile model that is used in the creation of ICE-5G (Peltier 2004). As alternative models profiles we select models profiles that have been shown by Root et al (2015b) to provide a good fit to sea level data, GPS and GRACE data in Fennoscandia in Root et al. (2015b). That study found two model setsviscosity profiles, one with higher viscosities in upper and lower mantle viscosity and one with lower viscosities. The fact that sediment loading is not taken into account to obtain viscosity profiles in Root et al. (2015b) will have a minor effect given that three very different viscosity profiles are selected to account for uncertainty in viscosity. Out of those sets we select the two models M8-128-150 and M4-16-80, where the first number denotes the upper mantle viscosity in $10^{20}$ Pa s, the second number denotes the lower mantle viscosity in $10^{20}$ Pa s and the third number denotes the lithosphere thickness in km. The three viscosity profiles are shown in Fig. 1. Note the lower viscosity $10^{22}$ Pa s in VM5a just below the lithosphere, from 60 to 100 km depth.

[Figure]

**Figure 1: The three viscosity profiles used in this study. M04 refers to viscosity profile M4-16-80, M08 refers to M8-128-150.**

Since we are only interested in the effect of GISR the exact ice loading history is of less importance, and only influences the effect of GISR through the distribution of meltwater possibly replaced by sediment, which is a smaller effect than the sediment loading itself. For ice loading history the ICE-5G modelv1.2 (Peltier 2004) is selected which is provided with . Ttime steps are of 2,000 years from 120 kyr Before Present (B.P.) to 32 kyr B.P., 1,000 years from 32 to 17 kyr B.P. and 500 years from 17 kyr B.P. to present.

**3.2 Model inputs: sediment distribution**

In order to model the loading effect of GISR it needs to be known how much sediment is transported, where it came from and where it is deposited. Erosion and deposit estimates for entire all of Scandinavia during the last glacial cycle are not readily available. Therefore, we created a map of sediment deposition from estimates in the literature of sediment volumes transported in smaller local features: through mouth fans (TMFs) (i), large scale failures (ii) and basin flux (iii). Each of the features will be briefly discussed in the following.

(i) TMFs are places where rapid flowing ice streams at the end of the continental shelf converge, and where sediment is deposited off the shelf, see Figure 2 for the locations. Local observations come from sonar and seismic profiling (Dowdeswell et al., 1996) and coring (Saettem et al., 1992; Laberg and Vorren 1996; Taylor et al., 2002; Laberg et al., 2012). Other estimates come from modelling based on bathymetry, elevation and environmental conditions (Siegert and Dowdeswell 2002), but these are highly dependent on the amount of ice that is believed to have existed in the Barents and Kara Sea. Deposition also takes place on the shelf, but is probably smaller (Zieba et al. 2016)

The estimates are compiled in Table A1 in Appendix A.

(ii) Large scale failures represent the sediment that is displaced after collapse of the slope. The largest of such events related to the Eurasian ice sheet is the Storegga slide. Haflidason et al. (2004; 2005) estimate the Holocene event to have a volume

of 2400-3200 km$^3$ based on sonar scans and sedimentary cores. A compilation of the studies for this and other slides is provided in Table A2 in Appendix A.

[Figure]

**Figure 2: Schematic representation of the locations of Through Mouth Fans (purple), Large scale failures (brown) and basins (green).**

(iii) The TMFs and the large-scale failure are examples of local features. However, constant deposition of sediment from the source area to the basins () also takes place. During glaciations, sediment activity is increased, with thickness changes estimated to be 5 cm kyr$^{-1}$ and 2 cm kyr$^{-1}$ for the periods of 30-10 kyr B.P. and 10-0 kyr B.P. respectively (Taylor et al., 2002), which amounts to a total volume of 97 and 58 km$^3$ for the Norwegian basin and 485 and 290 km$^3$ for the Lofoten basin. There are also channels and canyon systems that are not captured by either the large scale individual features which

still provide larger sedimentation rates than the overall basin estimate. Estimates for the Lofoten channel system amount to 35 km$^3$ (Taylor et al., 2000), which is small enough that it can be neglected compared to the other events. The basin fluxes are given in Table A3 in Appendix A.

5   Conflicting estimates are stated for GISR volumes in Tables A1-A3 in Appendix A, for example the Byørnøya Through Mouth Fan in Table A1. Also the timing is uncertain for most events. Therefore, different sets of GISR loads awere created, consisting of minimum, maximum and moderate estimates from the tables, which are labelled Sed1, Sed2 and Sed3, respectively. The time step is chosen to be the average of the time span given, rounded off to the nearest time step of the ice model. A value for sediment density of 2300 kg/m$^3$ is taken, in agreement with Amantov et al. (2008) and measurements for

10   shallow sediments (Zaborska et al. 2008). Uncertainty in the density is likely smaller than the uncertainty in thickness and timing which are represented here by the different sediment models.

[revised manuscript text omitted]

**Table 1: maximum effect of sediment loading for different GISR estimates. In each cell the three numbers separated by | correspond to earth models M4-16-80/M8-128-150/VM5a, respectively. The maximum effect on relative sea level measurements is calculated as the maximum effect at any of the 6 sites of Figure 3 at the time at which there are measurements (shown with vertical bars in Figure 4). The uplift rate is interpolated at the GPS sites of the BIFROST network presented in Lidberg et al. (2010) and the maximum is shown. The maximum positive rate of change of gravitational acceleration  in Scandinavia is determined in the land area contained in the box with longitudes from 5° to 37° and latitudes from 55° N to 71° N, see Error! Reference source not found.. The maximum positive gravity rate in the Barents Sea is determined in the box with longitudes between 10° and 100° degrees, and latitudes between 71° N and 81° N.**

[Figure]

**Figure 3: Colors denote the difference between RSL computed at LGM (26,000 years B.P. in the ICE-5G model) and computed at present caused by GISR for sediment model Amantov (left) and Sed1 (right). Note the different color scales. Numbered black dots show the locations of Relative Sea Level data used in Fig. 7. Earth model M4-16-80 is used for both cases.**

[Figure]

**Figure 4: RSL at selected sites from the Tushingham and Peltier (1992) database, for the ICE-5G model (red  line) and the ICE-5G model with sediment transport (blue line) according to the Amantov model in combination with the M4-16-80 Earth model. The brown line shows the difference, with scale on the right y-axis.**

While the relative sea level over time also includes the geoid effect due to removed mass, the present-day uplift rate only represents the viscous readjustment due to past changes in surface loadings. The pattern of uplift rates is shown in Figure 5 for the Amantov and Sed1 GISR models. In Figure 5a uplift can be seen in the formerly glaciated region of Scandinavia and subsidence off-shore, corresponding to . Figure 5b mainly shows the effect of large scale failures. In both figures, the largest effects are off-shore where no GPS measurements can be made. To see the effect on observed uplift rates, the second data column in Table 1 shows the maximum effect of the sediment loading at any of the BIFROST sites of Lidberg et al. (2010). GISR is seen to always increase uplift rates, because most of the GPS measurement stations are in previously glaciated areas from where erosion took place. Thus, when GPS is used to draw conclusions on or validate glaciation history (e.g. Kierulf et al. 2014) the contribution of the ice may be overestimated. .

[Figure]

**Figure 5: Uplift rates caused by sediment redistribution according to GISR models Amantov2 (left) and Sed1 (right). Dots denote locations of GPS sites of Lidberg et al. (2010). Note the different color scales. Earth model M4-16-80 is used for both cases.**

Finally, the influence on gravity rates is also investigated (Figure 6) , as gravity rates derived from the GRACE satellite mission constrain GIA in Scandinavia (Steffen et al., 2008; van der Wal et al., 2011) and the Barents Sea (Root et al., 2015a). To compare with GRACE data a maximum spherical harmonic degree of 60 is used in the GIA model, which is the same truncation used in many GRACE studies. Similar to the uplift rate signal, Sed1 reflects the signal of the landslides off-shore southern Norway and the Amantov model has negative gravity rates west of the Barents Sea where sediment is deposited. To evaluate the magnitude of the effect, Table 1 provides the maximum gravity rate that occurs in the areas that are used for GIA studies, Scandinavia and the Barents Sea, see  Figure 6. The values can be judged by comparing against the GRACE measurements error. The measurement error of the gravity rates derived from GRACE is computed using the method of Wahr et al. (2004), assuming that residuals obtained after fitting a trend, secular and annual signal to the monthly gravity fields reflect noise. This method was shown to results in a similar error magnitude as calibrated standard deviations or a full variance-covariance matrix (van der Wal et al., 2010). The measurement error propagated to the trend has a value of 0.016 µGal yr$^{-1}$ for a ten-year GRACE time series from January 2003 to July 2013. In Scandinavia sediment loading results in gravity rates  at the level of the measurement error for the Sed/2/3 models for the three earth models (between 0.002 and 0.014 µGal yr$^{-1}$), and larger effects for the Amantov model (between 0.007 and 0.027 µGal yr$^{-1}$). However, gravity rates derived from GRACE are not only affected by

measurement error. Using data from different processing centers, and using different correction models results in a larger spread in the gravity rate than the measurement error. Based on a comparison with GPS data the RMS error was estimated to be 1 mm/year in terms of uplift rate (van der Wal et al. 2011), which is around 10% of the maximum uplift rates. Assuming most of this reflects GRACE errors, this means that the gravity rate error is roughly 0.1 microGal/year. In this light, the

5 values in Table 1 appear to be insignificant, but the effect of sediment loading is systematic.  Note that the direct attraction of the current rate of sedimentation in the Barents Sea is not included in our computations. The effect of  current sediment transport could become relevant for GRACE studies and should be investigated in future studies.

10 .

[Figure]

**Figure 6:** Gravity rate due to GISR according to the Amantov model for viscosity profile M4-16-80. Boxes in Scandinavia and the Barents Sea denote the areas for which the maximum gravity rate is given in **Table 1**.

**5. Discussion and conclusions**

15 We investigated the effect of sediment transport during the past glaciation in Scandinavia on current GIA observables. The effect on present-day GIA observables is small compared to the effect of ice loads that are displaced during glaciation. Sediment uptake takes place over a large area, and deposition takes place in limited areas mostly confined to the ocean which can lead to locally higher signal near areas of sediment deposition. It was found that RSL data are not

20 significantly affecteds by GISR because those data are located  near the shore, in between zones of erosion and deposition, and because a large part of the deposition takes place early in the deglaciation when the errors in RSL data are relatively large. At the LGM the effect is 18 m; at 6 kyear B.P. the effect is below 2 m, comparable to what was found for the Indus River basin (Ferrier et al. 2015).

Also, the effect on present-day uplift rate and gravity rates is limited; depending on the estimate for sediment transport that is used, the magnitude of GISR loading effects is near the measurement limit, several tenths of mm yr$^{-1}$ uplift rate and several tenths of μGal yr$^{-1}$ gravity rate. The magnitude is comparable to a recent estimate of subsidence in the Mississipi delta (Yu et al. 2012; Wolstencroft et al. 2014) but somewhat smaller because of the larger sediment deposition area for the Fennoscandian ice sheet, and the earlier demise of the ice sheet in the Barents Sea. The magnitude is smaller than a possible reference frame bias in GPS derived uplift rates, and in the presence of other GIA model uncertainties several tenths of mm/year does not appear to be significant. However Nevertheless, the effect is systematic, reducing uplift rates and gravity rates in the land areas of Fennoscandia and Nova ZemblaSvalbard and increasing gravity rates west of Fennoscandia and the Barents Sea. Thus, if uplift or gravity rates are used to infer viscosity profiles or ice thickness those estimates could be biased. Also, a few tenths of mm/year is not negligible compared to global average sea level change.

[revised manuscript text omitted]

**Appendix A: estimates of sediment displacement**

| Through Mouth Fan | Volume [km³] | Timespan [ka B.P.] | Reference |
|---|---|---|---|
| Bellsund | -- | - | - |
| Byørnøya | 820-1100 | 22-17 | Taylor et al. (2002) |
| | 1360 | 21.5-17.5 | Laberg et al. (2012) |
| | 2700 | 30-8 | Siegert and Dowdeswell (2002) |
| | 4176 | Late Weichselian | Laberg and Vorren (1996) |
| | 4800 | 27-14 | Dowdeswell and Siegert (1999) |
| Franz Victoria | 500 | 30-8 | Siegert and Dowdeswell (2002) |
| Isfjorden | 22.5 | 30-0 | Elverhoi et al. (1995) |
| Kongsfjorden | - | - | - |
| North Sea | 800 | 30-0 | Taylor et al. (2002) |
| Storfjorden | 250 | 30-0 | Siegert and Dowdeswell (1999) |
| | 800 | 27-14 | Dowdeswell and Siegert (1999) |
| Svyataya Anna | Little | 30-8 | Siegert and Dowdeswell (1999) |
| | 2200 | 27-14 | Dowdeswell and Siegert (1999) |
| Voronin | little | 30-8 | Siegert and Dowdeswell (1999) |

Table A2: estimates for volume of sediment displaced in different events, timespan and reference. No estimates for the Bellsund and Kongsfjorden are available, but sediment transport there is expected to be relatively small.

| Large Scale Failure | Volume [km³] | Timespan [ka B.P.] | Source |
|---|---|---|---|
| Andøya | 900 | 11-0 | Taylor et al. (2002) |
| Byørnøyenna | 1350 | 20-15 | Leynaud et al. (2009) |
| Hinlopen | 1200-1350 | 30 | Winkelmann et al. (2008) |
| North Faroes | 135-1700 | 9.85 | Taylor et al. (2002) |
| Storegga I | 3880 | 50-30 | Bugge et al. (1988) |
| Storegga II-III | 1700 | 8-6 | Bugge et al. (1988) |
| Storegga II-III | 2400-3200 | 7.25 | Haflidason et al. (2005) |
| Traenadjupet | 900-1900 | 4.2 | Taylor et al. (2002) |

Table A3: Estimates for volume of sediment displaced by large scale failures.

| | Rate | Volume | Timespan | Reference |
|---|---|---|---|---|
| | [cm\ka] | [km$^3$] | [ka B.P.] | |
| Norwegian basin | 2 | 58 | 10 | Taylor et al. (2002) |
| Lofoten basin | 2 | 97 | 10 | Taylor et al. (2002) |
| Norwegian basin | 5 | 290 | 20 | Taylor et al. (2002) |
| Lofoten basin | 5 | 485 | 20 | Taylor et al. (2002) |
| Norwegian channel | - | 35 | 30-0 | |

**Table A4: estimates for volume of sediment displaced in the Lofoten and Norwegian basins and channel systems.**

| | Sed3 (max) | Sed1 (min) | Sed2 (moderate) | |
|---|---|---|---|---|
| **TMF** | Volume | Volume | Volume | Time span |
| | [km$^3$] | [km$^3$] | [km$^3$] | [ka B.P.] |
| Byørnøya | 820 | 1360 | 4800 | 22-17 |
| North Sea | 800 | 800 | 800 | 30-0 |
| Storfjorden | 250 | 300 | 800 | 27-14 |
| Franz Victoria | 500 | 500 | 500 | 30-8 |
| Svyataya Anna | 100 | 250 | 2200 | 27-14 |
| **Large Scale Failure** | Volume | Volume | Volume | Time span |
| | [km$^3$] | [km$^3$] | [km$^3$] | [ka BP] |
| Andøya | 900 | 900 | 900 | 8 |
| Bjørnøyrenna | 1350 | 1350 | 1350 | 18 |
| North Faroes | 135 | 1400 | 1700 | 10 |
| Storegga | 1700 | 2400 | 3200 | 7 |
| Traenadjupet | 900 | 1900 | 1900 | 4 |
| Hinlopen | 1200 | 1275 | 1350 | 30 |
| **Basin fluxes** | Volume | Volume | Volume | Timespan |
| | [km$^3$] | [km$^3$] | [km$^3$] | [ka BP] |
| Norwegian basin (interglacial) | 58 | 58 | 58 | 9-0 |
| Lofoten basin (interglacial) | 97 | 97 | 97 | 9-0 |

| | | | | | |
|---|---|---|---|---|---|
| Norwegian basin (glacial) | 290 | 290 | 290 | 30-10 |
| Lofoten basin (glacial) | 485 | 485 | 485 | 30-10 |
| Channel | 35 | 35 | 35 | 30-0 |

**Table A5: sediment estimates created from observations and estimates of individual estimates in tables A1 to A3, as described in the main text. Sed2 and 3 are the maximum and minimum models, respectively.**

**Appendix B.1: creation of spatial pattern from reported sediment displacement features**

To create the spatial distribution of erosion event shapes were drawn in the QGIS software package to match the figures in Taylor et al. (2002) and Winkelmann et al. (2008), see figure B.1. The shape files were converted to raster files with a sediment height for each point in the 256x512 spatial grid using MATLAB and GDAL. Across the source and sink areas of the sediments a uniform sediment height change is assumed so that the volume matches the estimates in Table A4. The areal extent is the largest source of uncertainty. Note that in reality deposition depends on distance from the source and also the area of deposition changes when the continental shelf is flooded or exposed (see e.g. Ferrier et al. 2015). In this study we opted to address the largest sources of uncertainty by using two independent estimates of GISR and three different Earth viscosity profiles.

**Appendix B.2: creation of spatial pattern from reported sediment displacement features**

The model of Amantov et al. (2011) couples ice sheet growth and erosion and is constrained by sedimentary and seismologic observations. The model accounts for enhanced erosion due to ice streams and erodability of various sub-ice surfaces among others. Sediment removal and deposition is shown in figure 3.6 of Amantov et al. (2011) in equivalent water thickness. Because part of the data is proprietary, the data for the time series and for creating the figure is not available. The software package QGIS was used to georeference the image and digitize the load at hundreds of points between which triangular interpolation was performed (figure B.2). The original model of Amantov et al. (2011) conserved mass, but in converting the graphics to a grid of sediment thickness values mass conservation was lost. We opted not to enforce mass conservation, as doing so would require further assumptions which introduce uncertainty. As a result there is a total geoid shift of around 13 cm over the glaciation, which causes a loading effect that is small compared to sediment thickness changes. To obtain a time series of sediment thickness it is assumed that the temporal variation of sediment transport follows that of total ice mass change, based on the common assumption that erosion rate is proportional to sliding velocity (see Herman et al., 2011). Recently, erosion has been found to be proportional to sliding velocity squared (Herman et al. 2015), which would enhance erosion rate during periods of largest ice change compared to our erosion history.

The color scale in 3.6 of Amantov et al. (2011) appears to be saturated at 300 m, but in fact 300 m is the best guess by Amantov et al. (2011) for maximum sediment deposition thickness and it set as maximum deposition in the modelling (Aleksey Amantov, personal communication). Finally, the large-scale failures for Sed1 from table A.5, which are not included in the model of Amantov et al. (2011), are added from Table A2.

[Figure]

**Figure B.1: shape of erosion events, derived from Taylor et al. (2002) and Winkelmann et al. (2008).**

[Figure]

**Figure B.2: Points captured in QGIS from Amantov et al. (2011) with value of sediment load expressed in meters of water load. Values of the points are interpolated using splines.**